



# Transversely Isotropic Lower Crust of Variscan Central Europe imaged by Ambient Noise Tomography of the Bohemian Massif

Jiří Kvapil[1], Jaroslava Plomerová[1], Hana Kampfová Exnerová[1], Vladislav Babuška[1], György Hetényi[2], and the AlpArray Working Group[+]

[1]Institute of Geophysics, Czech Academy of Sciences, Boční II/1401, 141 31 Prague 4, Czech Republic
[2]Institute of Earth Sciences, University of Lausanne, Lausanne, Switzerland
[+]For further information regarding the team, please visit the link which appears at the end of the paper.

*Correspondence to*: Jiří Kvapil (j.kvapil@ig.cas.cz)

**Abstract.** Recent development of ambient noise tomography, in combination with increasing number of permanent seismic stations and dense networks of temporary stations operated during passive seismic experiments, provides a unique opportunity to build the first high-resolution 3-D shear wave velocity ($v_S$) model of the crust of the Bohemian Massif (BM). The velocity model with a cell size of 22 km is built by conventional two-step inversion approach from Rayleigh wave group velocity dispersion curves measured at more than 400 stations. The shear velocities within the upper crust of the BM are ~0.2 km s$^{-1}$ higher than those in its surroundings. The highest crustal velocities appear in its southern part, the Moldanubian unit. The model provides compelling evidence for a regional-scale of velocity distribution. The Cadomian part of the region has a thinner crust, while the crust assembled, or tectonically transformed in the Variscan period, is thicker. The sharp Moho discontinuity preserves traces of its dynamic development expressed in remnants of Variscan subductions imprinted in bands of crustal thickenings. A significant feature of the presented model is the velocity-drop interface (VDI) modelled in the lower part of the crust. We explain this feature by anisotropic fabric of the lower crust, which is characterized as vertical transverse isotropy with the low-velocity being the symmetry axis. The VDI is often interrupted around the boundaries of the crustal units, usually above locally increased velocities in the lowermost crust. Due to the NW-SE shortening of the crust and the late-Variscan strike-slip movements along the NE-SW oriented sutures preserved in the BM lithosphere, the anisotropic fabric of the lower crust was partly or fully erased along the boundaries of original microplates. These weakened zones accompanied by a velocity increase above the Moho, which indicate an extrusion of mantle rocks into the lower crust, can represent channels through which portions of subducted and later molten rocks have percolated upwards providing magma to subsequently form granitoid plutons.

## 1 Introduction

The Bohemian Massif (BM) represents the easternmost relic of the Variscan orogenic belt in central Europe . The massif formed approximately between 500 and 250 Ma as a collage of microplates and relics of magmatic arcs as a result of the large-scale collision of supercontinents Laurasia and Gondwana (e.g., Franke et al., 2017). The core of the BM consists of

three tectonic units - the Saxothuringian (ST), Teplá-Barrandian (TB) and Moldanubian (MD), which represent originally independent microplates. The eastern part of the BM consists of the Moravo-Silesian Zone with the Neoproterozoic Brunovistulian (BV) basement beneath its southern Moravian (MO) and northern Silesian (SI) parts (Fig. 1).

Early Controlled Source Seismic (CSS) research of the BM crust in the 1960s found the thickest crust of about 40 km in the MD unit and the thinnest crust (~27 km) in the western part of the BM, in the ST unit (e.g., Beránek et al., 1975). The initial research was followed by many refraction and reflection seismic experiments which provided additional data for modelling the BM crust. Karousová et al. (2012 and references therein) interpolated the compiled findings of wide-angle refraction and near-vertical reflection seismic profiles into the first three-dimensional P-wave velocity model of the BM. The model

maintains the gross features of the individual earlier models of the BM crust.

For a long time, the active seismic experiments have represented the only source of data for studies of the BM crust due to the lack of natural earthquakes needed for local earthquake tomography of the massif. Complementary approaches, particularly the receiver function (RF) method (e.g., Hetényi et al., 2018b) and the more recently introduced ambient noise

tomography (ANT) have been proved to be useful tools for imaging the crust. The first one analyses delay times of converted phases from teleseismic earthquakes and the second is based on cross-correlation technique (e.g., Bensen et al., 2007). The ANT has the ability to model the Moho and to retrieve shear wave velocities across the Earth crust and of the uppermost mantle when the network aperture is sufficiently large. One of the main advantages of the ambient noise tomography is in its ability to exploit long-term series of seismic ambient noise recorded at seismic stations regardless of

earthquakes occurrence and their spatial distribution. Resolution of velocity models depends prevailingly on station density, inter-station distance and azimuth variability and the overall path coverage of the studied region.

In this paper, we present the first homogeneous high-resolution 3-D shear velocity model of the whole BM crust, infer Moho depth and map its lateral variations.  The unprecedented high resolution of the velocity model is achieved thanks to data

from temporary stations of the large-scale AlpArray passive experiment (Hetényi et al., 2018a) and several regional passive seismic experiments organized in the BM and its surroundings during the last two decades (see below).

We introduce a semi-automated procedure for ambient noise tomography, which consists of data gathering, waveform pre-processing, calculation of station-pair cross correlations, the Rayleigh wave extraction, frequency-time analysis (FTAN),

discretization of the velocity dispersion curves of individual station pairs and cross-dimensional dispersion curve tomography. The procedure leads to the homogeneous 3-D velocity model constructed from 1-D velocity-depth functions at a dense grid. The stochastic inversion with the use of the Improved Neighbourhood Algorithm (Wathelet, 2008) provides several additional attributes, e.g., misfit, skewness, standard deviation, as well as estimates of uncertainty, allowing us to



assess reliability of the resulting velocity model. We compare the new 3-D model with several 2-D CSS profiles published
by other authors and with estimates of the Moho depths from RF, resolved within this study. Velocity decrease modelled in
the lower crust is explained by the vertical transverse isotropic (VTI) fabric with the 'slow' symmetry axis.

## 2 Data

To evaluate the shear velocities within the BM crust, we processed vertical components of daily recordings with sampling
rate from 20 to 200 Hz from 410 stations located in the massif and its broader surroundings framed at 7.1°–21.4° E and
46.2°–56.3° N and operated between 2002 and 2016. We used recordings from permanent observatories, temporary stations
of the AlpArray passive seismic experiment (AlpArray Seismic Network, 2015), its target-oriented complementary field
experiment EASI (Hetényi et al., 2018b; network doi: 10.12686/alparray/xt_2014), as well as recordings from previous
temporary measurements in the BM, e.g., BOHEMA I-IV (Plomerová et al., 2007; 2012; Karousová et al., 2013), PASSEQ
(Wilde-Piorko et al., 2008), and EgerRift (Babuška and Plomerová, 2013) (Fig. 2). Thorough analyses of noise sources
showed that 90-day-long interval in the summer season (from June to August) is the optimal time period for stacking cross-
correlation functions in the BM (Fig. 3). Strong microseismic energy arriving from the North Atlantic Ocean, particularly in
the winter, strongly disturb the ambient noise wave field in central Europe and thus corrupt the fundamental assumption of
the method to have the isotropic distribution of ambient noise sources. The summer season selection of data ensures
similarity of the causal and acausal parts of the CCF (Fig. 3c) and the isotropy of the signal-to-noise ratio (Fig. 3d).

For our study, we created 83 845 station pairs from all of the available data and kept only those which satisfy conditions as
follows: inter-station distance is between 50 km and 600 km, station-pair midpoint is inside the area framed at 11.3°–19.3° E
and 47.5°–52.4° N, and station pairs contain at least 61 days of simultaneous recordings. With these selection criteria, we
keep 24 258 good quality station-pair data suitable for ambient noise processing.

## 3 Methods

Theoretical framework for calculation of surface wave velocity dispersion curves from ambient noise was established in
many experimental (e.g., Ren et al, 2013; Kästle et al., 2018; Lu et al., 2018; Schippkus et al., 2018; Soergel et al., 2020; and
Guerin et al., 2020) and theoretical studies (e.g., Dunkin, 1965; Lobkis and Weaver, 2001; Levshin and Ritzwoller, 2001;
Sambridge, 1999; and Yoshizawa and Kennet, 2005). The studies are based on cross correlations of long sequences of
broadband seismic recordings and subsequent inversions of dispersion-curves for shear wave velocity model. Among others,
Lobkis and Weaver (2001) and Shapiro and Campillo (2004) showed that the cross correlations represent estimates of
Green's functions for the crust and uppermost mantle along paths connecting two arbitrary sites. The reliable Green's
functions can be retrieved, if the medium is homogeneous and isotropic and if the ambient noise wave field is diffuse or it
arrives from uniformly distributed noise sources. However in seismology, none of these assumptions is fully satisfied.
Therefore an additional processing is needed for modelling structure of the crust. Bensen et al. (2007) describe data
processing that delivers reliable surface-wave dispersion measurements, widely used in many recent studies mentioned
above, even if the above conditions are not fully satisfied. We have implemented the approach in building the first detailed
3-D velocity model of the BM. Thanks to significant dispersion of surface waves and to their frequency-dependent depth
penetration, the surface wave tomographic inversion can be solved in computationally efficient two-step approach. The two-
step method integrates an iterative fast marching method resulting in shear velocity maps per frequency bands and a
stochastic inversion of dispersion curves for a collection of multi-layered shear velocity models. The stochastic inversion
addresses a non-uniqueness of Dunkin's (1965) formulae by a search for the best fit between the modelled and measured
dispersion curves.

We build the 3-D shear wave velocity model in three semi-automated phases (Fig. 4) by joining three existing software
packages. The MSNoise package of python codes (Lecocq et al., 2014) is applied for data pre-processing, calculation of
cross correlations and their stacking in Phase 1. The package is complemented by a new set of Python codes for
frequency-time analysis (FTAN) of Levshin and Ritzwoller (2001). This step includes careful velocity-frequency picking of
the group velocity dispersion curves. The FMST package of the Fortran codes (Rawlinson, 2005) is used to solve the 2-D
surface wave travel time tomography in Phase 2. It employs eikonal equation forward modelling and the iterative subspace
method to minimise RMS error between measured and computed travel times (Rawlinson and Sambridge, 2005). The non-
linear stochastic inversion of dispersion curves (Wathelet, 2008) is solved in Phase 3 with the use of the Dinver software
package (Wathelet et al., 2020).It employs the Neighbourhood Algorithm (NA) as an improved inversion method to search
for the best fitting models (Sambridge, 1999; Wathelet et al., 2004; Wathelet, 2008).


In Phase 1, we demean daily seismic records, resample them to 0.1 s interval with anti-aliasing filter, remove instrument
responses, normalise amplitudes and apply spectral whitening. Then we compute daily cross-correlation functions (CCFs)
for all station pairs by averaging cross-correlations computed for one-hour time windows with 30 minutes overlaps. We
extract fundamental mode of the Rayleigh waveforms and apply commonly used frequency-time analysis (FTAN). The
FTAN diagrams (Fig 5a) display amplitudes of waveform envelopes as a function of the group velocity and the period of
narrow band-pass filters. On the FTAN diagrams we measure group velocities of the Rayleigh waves. The group velocity
dispersion curves are discretized efficiently by an automated picking with non-linear one-third-octave band segmentation of
the frequency axis in the FTAN diagrams.

The automated picker starts from the third sample before the long-period cut-off point and continues in direction towards
both the short and long periods. In the short-period band, the picker prioritizes the low velocity local maxima. This prevents
picking the higher modes, as well as skipping and mixing different phases at the shot-period range. The cut-off period is a





function of interstation distance and average group velocity, and relates to the separation of the causal and acausal parts of the Rayleigh wave on the CCF. By visual inspection of the FTAN diagrams, we reject 13 % of station pairs and end up with

21 066 dispersion curve measurements providing velocity-period pairs in a period range from 2 to 74 s (Fig 5b and see Fig.13a).

In Phase 2, we split whole set of station-pair dispersion curves and cluster the velocities according to the common periods. We convert measured group velocities into travel times and solve the travel time tomography by the Fast Marching Method

(Rawlinson, 2005). Residual travel time RMS errors converge to optimal velocity model after 6 iterations. The dense inter-station spacing allowed us to construct velocity maps at a grid of 0.1° E-W by 0.1° N-S. In the initial model, we use constant velocity of 3.5 km s$^{-1}$ and constant radius smoothing in order to avoid introduction of artificial anomalies. Due to the well-known lower resolution at longer periods we increase the size of the smoothing factor proportionally to each period. We construct Rayleigh wave group velocity maps for period range from 2 to 74 s in an area framed at 7°–15° E-W and 47°–54°

N-S.

In Phase 3, we assemble the velocity slices computed in Phase 2 and construct dispersion curves at a regular grid of 0.3° E-W by 0.2° N-S. Then we invert the regularised dispersion curves for local 1-D layered shear velocity models with the use of NA in the stochastic inversion program. The reliability of the NA solution depends mainly on the number of randomly

generated starting models, number of iterations and the model space. These parameters need to be carefully tested to guarantee the complete sampling of the solution space. The overall quality of the inversion results is assessed by misfit values, which indicate how each model fits the input dispersion curve by the normalised RMS error. We tune the NA parameters by evaluation of the misfit values, the skewness and standard deviations of the group of best-fitting models with the aim of achieving the Gaussian distribution of group of best-fitting solutions without a bias of the NA search constraints.


The NA models consist of six-layers over a half-space (Table 1). The top three layers encompass heterogeneous shallowest structures and the three deeper layers represent the upper part and lower part of the crust and the uppermost mantle. The NA searches for the best fitting model in 360 iterations starting from 280 randomly generated initial models. To get a representative model at each grid node, we average 10 % of the best-fitting models from the resulting set of 400 000 models.

We apply two approaches of averaging. First, the conventional one, which averages the $v_S$ over the fixed depths, results in a gradient-velocity model (Fig. 6, left, blue curve). The second approach averages over layer indexes ($v_i, z_i$) and results in a layered-velocity model (Fig 6a, left, red curves). This approach allows us to retrieve the velocity boundaries to which the surface waves are insensitive in principle. The dispersion curve, calculated from the layered-velocity model, matches better the input dispersion curve (Fig. 6a, right).




The averaging over layer indexes leads to the straightforward implementation of a layer-stripping technique into the stochastic inversion. The layer stripping appears to be efficient tool to enhance the NA search in deeper layers of the model, where the solution suffers from the significant decrease of sensitivity kernels with depth (Yoshizawa and Kennet, 2005). The recursive stripping of the upper layers increases depth sensitivity and leads to better fit at long period range of the dispersion

curves (Fig 6b). Figure 6e reveals the linear improvement of average misfits for two selected period intervals, after each of the three steps of the subsequent layer stripping. The misfit reduction is feasible and consistent for all nodes. Similar improvement concerns setting the inversion parameter that allows or does not allow the velocity decrease in the lower part of the crust. Figure 6b,c shows that the models, in which velocity decrease with depth is allowed, fit better the dispersion curve with a local minimum around the 20 s periods. In our data set, a large majority of the resulting models consistently prefer the

velocity decrease in the lower crust (see also Fig. 10).

We assess the sensitivity of our inversion approach to the sharpness of the Moho as an interface by adding an additional gradient layer into the final 1-D model (Fig.6d). In order to mimic the gradational Moho we insert the layer on top of the uppermost mantle layer (Table 1) and varied its thickness from 2 to 10 km. The test shows that the larger is the thickness of

the transitional layer; the worse is the misfit in the dispersion curves, which is detectable from layer thickness larger than 2 km. This means that in case of the gradational Moho instead of a sharp discontinuity, the thickness of the transitional layer does not exceed 2 km.

## 4 Results

The presented high-resolution model shows shear wave velocities ~0.2 km s$^{-1}$ higher in almost the entire BM upper crust, on

average, than those in its surroundings, with the highest velocities in the southern part of the Moldanubian (MD) unit (Fig 7). The higher velocities clearly contour the BM shape. Very low velocities down to 10 km depth mark the SE rim of the BM beneath the Carpathian Foredeep. At greater depths (15 km), the relatively higher velocities are found in the MD unit. Locally increased velocities occur in the lower crust at several places, e.g. beneath the Eger Rift at 25 km depth. The upper mantle velocities arise first at 28 km depth slice beneath the Eger Rift (ER), marking thus the massif thinning along the rift.

While we find the top of the upper mantle at 32 km in the Saxothuringian (ST) unit, it is as deep as 39 km in the MD unit.

Figure 8 shows depths of the Moho discontinuity, as an interface, across which the lower crust velocities change into the upper mantle velocities. The crust of the BM is thinnest in its Saxothuringian part, beneath which the upper mantle velocities dominate at ~ 32 km depth, with very thin crust beneath the ER (~28 km). The ambient noise tomography detects the

thickest crust beneath the Moldanubian part of the BM, where the Moho deepens at least to ~38 km. The Moho as deep as 36 km has been detected in a small area around the boundary of the West Sudetes and the Lugian Domain around the Intra-Sudetic Fault (ISF).





Regardless of the shallowest, not well resolved upper 3 km, the model comprises two well-resolved layers, which we relate
to the upper and lower parts of the crust (UPC and LPC in Fig. 8b,c,). In the UPC the average velocities vary from ~3.2 km
s$^{-1}$ in the north and north-west of the massif up to more than 3.5 km s$^{-1}$ in the southernmost part of the BM. With two
small-size exceptions (along BOG and Lugian/Moravo-Silesian contact), the LPC of the BM can be divided into the
south-western and north-eastern half, exhibiting relatively high and low average velocities, respectively.

Individual 1-D velocity-depth models resulting from the stochastic inversions and their corresponding dispersion curves
exhibit distinct regional variations of the velocity interface between the LPC and UPC, which are characteristic for tectonic
units of the BM (Fig. 9a,b). Comparison of 1-D models with the radial Earth model IASP'91 (Kennet and Engdahl, 1991)
shows that the shear velocities retrieved in the upper part of the crust are mostly similar, while in the lower part of the crust
the ANT model velocities are significantly lower.


Most of the massif is characterized by a velocity decrease in the lower crust, which can be quantified as $dv_S = v_S(LPC_1) - v_S(UPC_7)$, where indexes denote respective sub-layers (Table 1). The velocity-drop interface (VDI) with negative $dv_S$ occurs
at variable depths and with different size of the velocity drop (Fig. 10). Within the MD, the interface with velocity drop
exceeding -0.1 km s$^{-1}$ lies at the 26-30 km depth, being the deepest in the south, though the average velocity decrease is not
the largest one (14.7° E 49.5° N, depth=28 km, $dv_S$=-0.18 km s$^{-1}$). The MD is surrounded by a band, in which the VDI uplifts
into shallower depths. In the southwest continuation of the MD unit out of the BM(see Fig1), the size of the velocity
decrease become even stronger locally, but the depth of the interface shallows by ~10km (11.7° E 48.7° N, depth=19 km,
$dv_S$=-0.37 km s$^{-1}$ ). The amplitude of the negative $dv_S$ slightly increases also towards the TB unit, but its depth remains
deeper than that in the westward continuation of the MD unit out of the BM (13.5° E 50.1° N, depth=23 km, $dv_S$=-0.27 km s$^{-1}$
$^{1}$). The SW-NE trend of the Variscan structures, including the ER, is reflected in a separation of the ST, characterized by the
weaker velocity decrease across the VDI at shallower depths (12.0° E 50.7° N, depth ~18 km, $dv_S$=-0.21 km s$^{-1}$) in
comparison with the rest of the BM. In the south-western part of the region (11.4° E 49.3° N), where the ST unit neighbours
the crust of the MD zone (out of the BM), the VDI is absent. The narrow zone without the velocity drop continues
north-eastward along the ER (ST/TB boundary). Further to NE, the ER continuation approximately divides the regions with
the shallow and deep VDI in the West Sudetes and Lugian Domain, respectively. The VDI is a little bit more expressed
around the Elbe Fault Zone (EFZ) and lies deeper to the north-east of the EFZ in comparison with the ST south-west of the
EFZ, but it remains shallower in comparison with the MD. Further to the east in the northern BM, the VDI deepens to ~25
km, in the northern Lugian Domain and shallows again to ~20 km in the Silesian (Fig. 10).

Besides the tiny, but systematic differences between average velocities in the upper and lower crust (see Fig. 7b,c,), there are
also differences in the velocity gradients in the individual regions of the BM (Fig. 10). The EFZ divides the BM





approximately into two zones. The southern zone of the BM, which includes the MD and TB units, the eastern ST unit and southern part of the Lugian Domain, is characterized by higher velocity gradients below the VDI than that in the layer above the VDI. In the north-eastern rim of the BM, particularly north of the ISF, in the East Sudetes and Silesian Domain, the

gradient difference at VDI is negative, i.e., the gradient in the layer above the VDI is higher than that below the interface. The differences between velocity gradients in the UPC and LPC clearly delineate zones of the BM that match with the BM zonation according to the size and depth of the velocity drop at the VDI (Fig. 10).

Distinct lateral variations of the $v_S$ velocities, the VDI and the Moho depths, as well as thickness of the LPC are shown along

several cross sections in the 3-D view (Fig.11, for more views see S1–S4). The sharp dark blue-light green velocity contrast marks the Moho derived from the ambient noise tomography. The figures show crustal thinning, e.g., beneath the ER (see also Fig.15) and Moho undulations, e.g., around crossing point of Profile 7 and Profile 12 beneath the southern SI (eastern continuation of the EFZ). The shallow Moho beneath the ST unit is also visible along the Profile 14 (south-western part in view from SW).


The Ps receiver functions represent an independent method to estimate the Moho depth from teleseismic P-to-S converted phases at the bottom of the crust. We picked the Ps-P delay times for all the stations and calculated the Moho depth with the use of the IASP91 velocity model (for more details see Hetényi et al., 2018b). We interpolate the Moho depths estimates beneath individual stations without the back migration to true conversion points. In the core of the BM (the TB and the

south-western part of the MD units) the Moho depths from the time delay of the Ps converted phases ($Moho_{RF}$) and from the ambient noise tomography ($Moho_{ANT}$) are in agreement. In the remaining part of the BM and the Carpathian Foredeep, the $Moho_{ANT}$ is deeper, except of the very southern MD. The most frequent difference between the $Moho_{ANT}$ (Fig. 8a) and $Moho_{RF}$ (Fig. S5) attains 3km in the whole region (Figs. S6-7). The Moho depth estimated from the Ps-P delay times with the use of the average crustal $v_S$ from our ANT model (vANT) and standard $v_P/v_S = 1.729$ is even shallower. The most

frequent value attains almost 4 km. This reflects the velocity decrease in the LPC in the most of the BM velocity model, which shallows the Moho estimates from RF, on average, if standard $v_P/v_S$ from the IASP'91 is used. Aiming at attaining the same depth of Moho as in the ANT model ($Moho_{ANT}$) requires $v_P/v_S$ 0.15 lower than that corresponding to the IASP'91 model or $v_P/v_S$ from Zhu and Kanamori method (2000) retrieved from the same data set. There are clear regional variations (S8) which relates mostly to velocity variations in the LPC.


Mostly in the northern and eastern parts of the BM the Moho depths differ more than would correspond to the estimated accuracy of +/-2 km (e.g., Fig. 11, Profiles10, 14 and 15, and S1-4). Exceptional large discrepancies occur in regions southeast of the BM with uncompensated sediments in $Moho_{RF}$ calculations. Lower average velocities in the ANT model shallow the Moho depth estimates from the RF by 1 km, relative to that if the IASP'91 velocities are used, on average.






The most distinct phenomenon of the presented model is the detection of the velocity-drop interface (VDI) in the lower crust with the velocity decrease exceeding -0.1 km s$^{-1}$ (marked by yellow lines in Fig. 11). Variations of both the interface sharpness (size of the velocity drop) and its depth change laterally. These changes can be related to individual tectonic units of the BM crust (see Figs. 9 and 10). On the other hand, there is no direct correlation between the Moho, or the VDI depths,
and the lower crust thickness. Though the thin and thick lower parts of the crust beneath the VDI are visible above the deep and shallow Moho, respectively (e.g. P16, view from the W), the reversed relation, i.e., the thin and thick LPC above the shallower and deeper Moho, respectively, can be seen as well (e.g., P10, view from the W, or P14 from the E). This may reflect the fact that the accuracy of determining both the Moho and VDI depths is not sufficient to quantify the subtle relationship between the thickness of LPC and Moho depth.

## 270    5 Resolution and Sensitivity Tests

We use standard synthetic checkerboard tests to assess the resolution of the fast marching tomographic inversions. We search for spatial resolution on checkerboard patterns with +/- 3 % velocity perturbation relative to the initial velocity model (3.5 km s$^{-1}$) and with three cell sizes (Fig. 12a). As expected, the spatial resolution relates to the ray-path coverage for each period. In the central and southern parts of the model, we retrieve velocity heterogeneities at size of 44 km by 44 km using a
ray-path coverage at period 6 s (Fig. 12a,b left), 66 km by 66 km at period 19 s (Fig. 12b central) and 88 km by 88 km at period 48 s (Fig. 12b right). The resolution decreases towards the northern part of the model and towards the model margins with sparse ray-path coverage. The ray-path coverage of the region is shown in Fig. 12c by the ray-path hitcount per 0.3° E-W by 0.2° N-S cell size.

Further, we verify reliability of the fast marching inversion at different periods based on travel time residuals computed for all input travel paths through the updated 2-D velocity slices. The travel time residuals document that the FMST inversion returns reliable solution in a period range from 6 to 48 s (Fig. 13b). The computed travel time residuals at short periods (<5 s) exceed significantly the short wavelengths due to increased uncertainty of dispersion curve picking. Therefore, the fast marching inversion does not provide reliable results at short periods. The lower reliability of the fast marching inversion at
long periods (60 and 74 s) is affected by reduced number of the travel time inputs into the inversion (Fig. 13a).

Finally, we also calculate the depth sensitivity kernels at each cell of the final velocity model and demonstrate their well-known dependence on period (Fig. 13b).  The average sensitivity kernels, calculated across cells in the BM, show the highest sensitivity to structures in the upper part of the crust (~3–20 km) for periods between 6–19 s. The lower part of the
crust down to Moho is sampled well with periods >19 s. Based on the resolution and sensitivity tests, we infer that the VDI - the new phenomenon retrieved in the presented model at depth ~15–30 km - lies in a high-fidelity area for both the sensitivity kernels and minimum travel time residuals. Results of the spatial resolution tests confirm that we can resolve





structures at least as small as ~40 km in the upper part of the crust and ~80 km in the lower crust in the velocity model from the ANT.

## 6 Discussion

Traditionally, the upper and lower crust are distinguished in velocity models of the Earth crust, by higher velocities in the lower layer and the strong positive velocity contrast at the bottom of the crust (Moho). The mid-crust boundary, called Conrad discontinuity, was originally supposed to separate the granitic and mafic crustal layers. However, suggested Conrad discontinuity is missing in many seismological models with gradient velocities. In this paper, we show that an intra-crust boundary can appear with a negative velocity contrast in the lower crust.

The dispersion curves of group velocity computed by FMST in the BM, are characterized by steep slope of velocity increase in short period range <8 s. The curves have flat local maxima between 10 and 15 s periods, followed by a subtle velocity decrease with local minima at ~20 s and moderate velocity increase for periods larger than 20 s. General level of the local maxima of the dispersion curves attains group velocities of ~3.1 km s$^{-1}$. This value is lower than velocities from steeply-increasing dispersion curves of Lu et al. (2018) in the BM. But, the general shape and velocity level of our dispersion curves are comparable with those of Ren et al. (2013), Shippkus et al. (2020), or Qorbani et al. (2020). Also the ANT models of all these authors show higher $v_S$ in the BM upper crust than in its surroundings.

Wide ranges of 1-D layer thicknesses and velocities in our ANT input parameters (see Table 1) results in the $v_S$ model of the BM crust with three characteristic depth zones. The uppermost part, thinner than 3 km, is not well resolved due to locally heterogeneous structures including sediments in some places. For the two main parts of the crust we use the same notation as introduced by Artemieva and Thybo (2013) in the European EUNA model derived from a compilation of the CSS models. The upper part of the crust (UPC) includes felsic material with P velocities of 5.6–6.4 km s$^{-1}$ and the middle crust with intermediate composition and velocities 6.4–6.8 km s$^{-1}$. The lower part of the crust (LPC) denotes the mafic lower crust with velocities 6.8–7.2 km s$^{-1}$ and the high-velocity lowermost layer with $v_P$ higher than 7.2 km s$^{-1}$.

However, contrary to the commonly observed P-velocity increase with depth, we retrieved lower shear velocities in the LPC than in the lower part of the UPC. No mid-crust layer or an interface with positive impedance (traditional Conrad discontinuity) is required in our gradient velocity model. The characteristic velocity-drop interface (VDI) between the UPC and LPC can be associated with a thermodynamically controlled interface or rheological boundary representing the modern view of the Conrad discontinuity (Lay and Wallace, 1995).





## 6.1 BM crust images in ANT and CSS

Several CSS profiles criss-cross the BM (Fig. 14) and allow us to compare characteristics of the velocity cross-sections
through the 3-D ANT $v_S$ model with the 2-D $v_P$ models from series of the CSS measurements. For that we superimpose the
$v_S$ velocity model with main features of the CSS profiles, especially with the Moho discontinuity and inter-crustal
boundaries.

Two parallel CSS profiles Sudetes S04 (Hrubcová et al., 2010) and Celebration CEL09 (Hrubcová et al., 2005) cut across the
Variscan structures in the BM. While the Moho depths in central parts along the CEL09 profile and the ANT cross-section
overlap (~150–300 km of the profile, Fig. 14b), the $Moho_{CSS}$ along the S04 is 6 km shallower than the $Moho_{ANT}$, (~100–300
km, Fig. 14a). The $Moho_{RF}$ follows the shallower $Moho_{CSS}$ relief there and remains shallower along the CEL09. The $Moho_{RF}$
could be associated there with the top of a layer with the higher velocities in the lowermost part of the LPC (Artemieva and
Thybo, 2013). The Moho-depth differences increase in the SE parts of both profiles (~400-500 km) due to sediments in the
Carpathian Foredeep. Laminated high-$v_P$ gradient layers, modelled in the north-western and south-eastern parts of the
CEL09, fit neither the sharp $Moho_{ANT}$, nor the $Moho_{RF}$. The RFs are primarily sensitive to a velocity contrast across the
interface, but locally higher or lower crustal velocities affect accordingly the depth migrated Ps delay times. None of the
intra-crustal boundaries marked in the S04 or CEL09 profiles correlate with the distinct VDI in the ANT $v_S$ model.
Majdanski and Polkowski (2014) estimate uncertainties of velocities, as well as interfaces in the CSS models, and show
significant variations for deeper layers. According to their analysis, the $Moho_{CSS}$ lies in an interval of 9 km broad in the
marginal segments of the CEL09 profile, as a bottom of the high-gradient velocity layer with a velocity uncertainty of ~0.4
km s$^{-1}$. Even larger uncertainty of ~0.6 km s$^{-1}$ is estimated for P velocities modelled in the uppermost mantle.

Similar features exist in profiles S01 (Grad et al., 2008 ) and CEL10 (Hrubcová and Środa, 2008), both of which parallel the
NE-SW trend of the BM tectonics. Up to 5 km differences in the shallow $Moho_{CSS}$ and the $Moho_{ANT}$ exist northward from
the EFZ, locally being even larger for the $Moho_{RF}$. Two small high-velocity narrow blocks in the upper crust (~50–200 km,
S01, Fig 14c) are localized close to the higher velocities in the ANT model. Complex 2-D image of the Moho structure
modelled beneath these two high-velocity blocks in the S01 profile, is missing at the crossing point in the CEL09 profile
(Fig14b, ~100km), which indicates that an existence of the corresponding 3-D feature is improbable. The largest uncertainty
of $Moho_{CSS}$ depth of ~5 km is estimated beneath these two upper crustal structures and it is high in the whole central part of
the profile (Majdanski and Polkowski, 2014). The high-velocity gradient zone of a large extent is modelled along the
CEL10. This zone correlates with the model of the laminated Moho at the crossing point with CEL09. But there is no
laminated Moho on S04 at crossing point with CEL10. Tests of sharpness of the Moho in the ANT model at the two crossing
points do not indicate laminated Moho thicker than 2 km (S9). Instead, the localized high-velocity lowermost crust occurs
around both crossing points (CEL09xCEL10 and CEL10xS04) in the 3-D ANT velocity model (see Fig. 14a, d).



The Moho depths from different data sets and the three independent techniques fit best along the N-S profile ALP01 (Brückl et al., 2007) and the AlpArray-EASI (Hetényi et al., 2018b) running through the BM (Fig. 14e). In general, discrepancies in the Moho depth determination of less than 5 km, which is the most frequent case for the BM, are considered as a good match

(Kästle et al., 2018). We found a good fit also with the two-layer model of the crust beneath the KHC station (13.58N 49.13N), located near the AlpArray-EASI, ALP01 (~150 km) and 9HR CSS profiles (Červený et al., 1977). The authors applied the spectral ratio method by Phinney (1964) on data of the first European stations providing the continuous broadband digital recordings since 1972 (Plešinger and Horálek, 1976). The preferred two-layer model, with constant velocities, sets the Moho at 35 km, which is in agreement both with CSS results and the ANT model presented here

(Table 2). However, the upper/lower crust boundary is 5 km deeper in our model. While the velocity gradient in the upper crust is constant in our model too, we found the 0.2 km s$^{-1}$ $v_S$ decrease at the VDI and relatively steep positive velocity gradient in the lower crust. The velocity remains lower even at its bottom ($v_S$ =3.6 km s$^{-1}$). Also the uppermost mantle velocities are lower in the ANT model than those derived from the dispersion curves of teleseismic Rayleigh waves. Waves modelled in Červený et al. (1977) arrived from back-azimuths of ~30°, which is close to the high-velocity directions derived

from body-wave propagation of regional earthquakes (Plomerová et al., 1984). On the other hand, the ANT accounts for dispersion curves from all directions and thus results in lower velocities. An intra-crustal boundary in the Alp01(Fig. 14d) follows the VDI interface similarly to that in the CEL10, where it coincides with the top of the positive high-$v_P$-gradient transitional layer. In this case, the results from the trial and error technique of the 2-D modelling of the wide-angle CSS data differ from results obtained from the 3-D ANT, which model relatively low $v_S$ velocities in the LPC and the sharp Moho.


When comparing results from the CSS and 3-D ANT, one has to be bear in mind, that the 2-D CSS models are significantly affected by 2-D processing assumptions and narrow azimuth data acquisition, whilst the 3-D ANT model is based on 3-D homogeneous sampling and imaging the crust, but with relatively lower resolution in the lower crust. Therefore, the ANT maps 3-D structures of the crust in a more realistic way. Thanks to the layer-stripping method the ANT reveals the Moho as

a sharp discontinuity beneath the whole BM. However, the mantle velocities beneath the Moho remain lower than expected. Nevertheless, the lower velocities are in agreement with ANT of Ren et al.(2013), or results of teleseismic body-wave tomography (e.g., Amaru, 2007; Plomerová et al., 2016 and reference there in), as well as with full wave-form inversion (Fichtner and Villasenor, 2015). The shear velocities in the upper crust are in agreement with the $v_P$ CSS models considering the standard $v_P/v_S$ ratio. An apparent inconsistency occurs in the LPC, where the ANT tomography models a decrease of $v_S$

from the dispersive group Rayleigh waves, while the CSS models provide increasing phase velocities of longitudinal waves with depth. Sub-horizontal directions of propagation characterise both the Rayleigh and longitudinal waves, used in the CSS refraction or wide angle reflection exploration. In case of longitudinal waves, particles move in directions of propagation and thus sample horizontal velocities. On the other hand, the Rayleigh waves are polarized in vertical plane and the particles move along ellipses with longer vertical axis. Thus the Rayleigh waves sample a mixture of velocities, in which velocities in

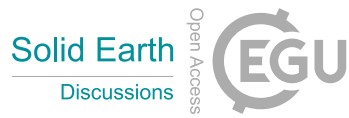

vertical directions prevail. Therefore, we can explain the discrepancy between the CSS and our ANT model by anisotropic structure (transverse isotropy with slow vertical symmetry axis) in the LPC (see the text below). Weak VTI anisotropy explaining the velocity drop in the lower crust is in agreement with results of other authors elsewhere (e.g., Luo et al., 2013; Shapiro et al., 2004; and Qiao et al., 2018). The subtle $v_S$ decrease in the top of the LPC of the ANT model (see Figs. 8 and 10) can be masked by uncertainties of the trial and error technique of the CSS models (Majdanski and Polkowski, 2014).


Some recent ANT models of the European crust (Lu et al., 2020), or the Alpine region (Kästle et al., 2018), represent themselves as candidates to a new reference model of the crust and uppermost mantle for the RF migration purposes. We show (see Section4 and S8) that in regions, where the $v_S$ ANT model detects low velocities in the LPC due to anisotropy, the average $v_P/v_S$ ratio is not valid for the simple $v_S$ conversion to $v_P$ in the crust.

### 6.2 Dynamic development of the BM crust


We aim at improving the current knowledge of tectonic processes that formed the BM by linking the most significant features of our $v_S$ velocity model of the crust with the near-surface geology exploiting numerous geological, petrological and geochemical studies focusing on the upper crust (e.g., Žák et al., 2014 for review). Commonly used statements about enigmatic structure and composition of the continental lower crust (e.g., Hacker et al., 2015), result from interpretations of

data from different seismic methods (Thybo et al., 2013 and references therein) which are often inconsistent. . Due to weak seismicity in the BM crust we know more about structure and composition of the BM mantle lithosphere (e.g., Plomerová et al., 2012; Ackermann et al., 2016) than about its lower crust.

#### 6.2.1 Traces of Variscan subductions

Crustal thickness varies between ~ 28 and 41 km in the ANT model of the BM. The relatively thin crust (<33 km) forms the

north- and south- western part of the massif (see Fig. 8a). The region with the thin crust is bounded by the EFZ in the north and by a 33 km depth band paralleling the ER in the east. The thinnest crust, where the Moho shallows up to 28 km, occurs near the intersection of the ER and its hidden north-eastern continuation east of the EFZ. The strike-slip EFZ is the boundary between the two unrelated parts of the ST unit (see Fig. 1), which obviously played an important role in the BM tectonics.

Transitions between the relatively thinner and thicker crust in the central and south-eastern BM spatially do not coincide with the surface tectonic boundaries of the TB unit. The thinner crust in the north-western part of the TB unit is characteristic for the older (Cadomian) part of the BM. To explain the crust thickening beneath the south-eastern part of the TB unit, we suggest the thrusting of the MD crust and the whole MD lithosphere under the TB unit (Babuška and Plomerová, 2013). The age of the ST/TB collision is estimated at ~380 Ma (Franke et al., 2017). On the other hand, the thicker MD and BV units in

the south-eastern part of the BM, were accreted to the TB block during the Variscan orogeny (e.g., Finger et al., 2007). Pitra et al. (1999) has already pointed out that there is no conformity of the TB ductile structures with those of the MD crust. The



Cambrian and Carboniferous tectono-metamorphic histories of the TB and MD, respectively, are separated by nearly 200 Ma. This age difference is reflected also in the difference between thicknesses of the crust.

Three discontinuous belts with thickness exceeding 38 km are mapped in the MD part of the BM (Fig. 8a). We explain the crust thickening most probably by remnants of three Variscan paleo-subductions. Two of them are in the MD adjacent to the two plutons the CBPC and the MDB. The third one relates to the subduction of the Brunovistulian lithosphere in the south-eastern rim of the MD. These processes that enlarged the pre-Variscan assembly of the ST and TB units, started by westward subductions of smaller oceanic plates beneath the TB block (Konopásek et al., 2002; Medaris et al., 2006; Finger

et al., 2007). Each of the oceanic subductions was followed by the continent-continent collisions. Thrusting of the two MD domains beneath the TB block was followed by under-thrusting of the BV domain beneath the MD continental lithosphere (see also Fig. 6 in Babuška and Plomerová, 2013).

Similarly, the thick crust in the Lugian Domain (Fig. 8) can represent a remnant of a small microplate that Konopásek et al.

(2019) identified by isotopic dating of monazite and garnet in high-pressure metamorphic rocks exposed in the northern part of the ST paleo-suture.  The Cadomian ST/TB suture, south-west of the EFZ, was later reactivated as the Cenozoic Eger Rift (ER, Fig. 1), while its north-eastern part, running along the northern part of the Lugian Domain, remained almost featureless on the surface. Nevertheless, Konopásek et. al. (2019) point out that existing tectonic models of the northern BM assume termination of the ST oceanic subduction at ~380-375 Ma by a collision and subsequent subduction of the ST continental

microplate. This might represent a record of a subduction and collision of a small island-like continental block, dated at ~340-337 Ma. These Variscan ages represent the youngest record of the ST continental subduction and the related high-pressure metamorphism reflected in the thickened crust of the ST unit eastward of the EFZ. In general, the thin crust of the western BM was consolidated during the Cambrian and Devonian times. On the other hand, the region with the thick crust witnessed Carboniferous processes, characterized by oceanic subductions and amalgamation of continental microplates

in the south-central and eastern BM.

The Moho map of Lu et al. (2018) constrained by a Bayesian inversion approach images two regions of thickened crust in the BM.  One of them can be related with the three belts of thickened crust in the MD unit (see Fig. 8a). Lack of separation of these small-size features  can be explained by lower resolution of 1.8° N-S in Lu et al. (2018)  compared with resolution

0.8° N-S in our model at Moho depth. Similarly, the ANT Moho depth of Kästle et al. (2018) shows low-resolution appearance of thicker MD crust without small-scale features due to its resolution of 2° N-S.

Variable thickness of the BM crust is the result of a dynamic phenomenon, in which remnants of Variscan subductions are reflected in the crust thickening. During the plate-tectonic processes, namely during the wandering of continental

lithospheric plates and their amalgamation into new continents, the deep crust and the Moho were continuously altered by





the strong rigid mantle lithosphere (Burov and Watts, 2006; Heron et al., 2016), the major stress guide in plate tectonic processes. In geologic time scale, the Moho is a dynamic, continuously modified feature. The images that we obtain are snapshots of the Moho depth at the present time, which preserve actions of tectonic processes involved in development of the crust over geological timescales (Carbonell et al., 2013).

**6.2.2 Anisotropy of lower crust**

The dominant feature of the presented 3-D $v_S$ model is the prevailingly low velocity in the lower crust (see Figs. 7, 9, 10 or 5bc), relative to the upper crust velocity. This is in contradiction with the high $v_P$ in the lower BM crust modelled by the CSS profiles. Velocities $v_P$ as high as 6.90–7.50 km s$^{-1}$ were modelled at depths 25–35 km in the ST crust and 6.60–6.95 km s$^{-1}$ in the TB and MD units (Profile CEL 09, Hrubcová et al, 2005). Such high $v_P$ velocities are close or even higher than those
interpreted in the lower crust of cratons (Mooney et al., 2002). Thybo et al. (2013) note that the top of the high P velocity lower crust might be mistakenly interpreted as the Moho, because (1) the high velocity lower crust may be a "hidden layer" for refraction seismic interpretations, (2) it may cause the strongest observed wide-angle reflection in seismic sections, or (3) it may be the strongest convertor in receiver function images.  We explain the difference between the high $v_P$ in the BM lower crust derived from the CSS and the velocity drop mapped by the $v_S$ velocities derived from the Rayleigh waves from
ambient noise by anisotropic structure of the lower crust. Refracted and/or wide-angle reflected P waves in the CSS sample prevailingly high velocities, while the Rayleigh waves sample the low velocities in the anisotropic medium-transversely isotropic with vertical 'low-velocity' symmetry axis.

The anisotropy of the lower crust plays an important role in velocity modelling and may explain some inconsistencies in
isotropic models of the crust and namely why the $v_P$ derived from the CSS profiles is high and on the other hand, the $v_S$ from the Rayleigh waves is low in the same crustal layer.  Several $v_S$ models incorporating the AlpArray data (Lu et al., 2018, 2020; Kästle et al., 2018; Schippkus et al., 2018; Guerin et al., 2020) have been derived from the ambient noise recorded at European stations. However, their resolution in the lower crust is often low outside of the Alpine region due to sparse station coverage in their ANT studies. Nevertheless, no velocity decrease in the lower crust that we observe in the BM, is mapped
either in their well resolved regions. On the other hand, a low-velocity zone in the mid-lower crust, representing a mechanically weak zone channelling material transport, was observed in the SE Tibet (Qiao et al., 2018). Also Luo et al. (2013) suggest an anisotropy in the lower crust beneath the Dabie orogenic belt in-between two cratons in central China, based on their ANT study. The fabric of the lower crust is associated with dynamic processes of continent-continent collisions and post-collision reworking. The only velocity inversion with depth beneath the BM is indicated in the $v_S$ model
of Schippkus et al. (2018), but their lower velocities are located in ~10–20 km depth in the upper crust.

To estimate maximum strength of the lower crust anisotropy, we assume a similar composition around the upper/lower crust boundary and associate the velocity decrease with the change of fabric. The velocity at the bottom of the upper crust can be





considered as average velocity of isotropic rocks or non-oriented material at the corresponding depth. Then the velocity
decrease at the top of the lower crust can be estimated as the velocity difference in the vertical and horizontal directions of
the anisotropic lower crust, approximated by the sub-horizontally oriented high velocities of the VTI. Then the minimum
strength of the lower crust anisotropy ranges from 3 to 9 % depending on the average $v_S$.

We have looked for rocks that could be the most suitable candidates for composition of the lower crust, namely as to
velocities and fabrics expressed in this type of seismic anisotropy. The lower crust represented by the Neukirchen-Kdyně
Massif (NK in Fig. 1 ) in the south-western tip of the TB unit, was thrust over the MD unit during their Upper Devonian
collision (Buess et al., 2002). The massif, where amphibolite-facies rocks are exhumed (Cháb and Žáček, 1994), can serve as
a model for a composition of the lower crust of the BM. Amphibolite, consisting mainly of hornblende and plagioclase, is a
most frequent candidate that corresponds best to interpretations in papers focused on seismic and gravity modelling of the
continental lower crust. Mafic granulite and gabbro are also considered in literature as potential components. However, both
rocks have typically very low $v_P$ and $v_S$ anisotropy measured in laboratory at high hydrostatic pressure (Almqvist and
Mainprice, 2017).

Ji et al. (2013) measured $v_P$ and $v_S$ velocities and anisotropy of 17 amphibole-rich rock samples at hydrostatic pressures up to
650 MPa corresponding to depths of about 20 km. Preferred orientation of hornblende plays decisive role in the formation of
seismic anisotropy of amphibolite, whereas other minerals as plagioclase, pyroxene or garnet diminish the anisotropy. The
authors found large variability of $v_P$ velocities (6.4–7.5 km s$^{-1}$) and concentration of very high velocities along foliation
planes (7.3–7.5 km s$^{-1}$). Similar high velocities were locally interpreted on the CSS profiles, e.g., within the lowermost crust
of the ST unit (Profile CEL 09, Hrubcová et al., 2005). Shear wave velocities $v_S$ measured by Ji et al. (2013) are also very
high (3.7–4.25 km s$^{-1}$), the lowest velocities (3.7–3.8 km) being for waves propagating perpendicularly to the foliation of
amphibolite samples. Such velocities correspond to the low-velocity propagations of Rayleigh waves in the vertically
transverse isotropic BM lower crust, considering Poisson solid and relating phase $v_S$ of body and surface wave velocities.

It is generally known that velocities measured in laboratory on fresh rock samples are usually higher than velocities
measured in large rock massifs that are heterogeneous and dissected by faults. The lower crust of the NK, taken as a model,
consists also of other rocks (meta-sediments, gabbro, see, e.g., Cháb and Žáček, 1994) that can decrease both the $v_S$ and
anisotropy.  On the other hand, an effective anisotropy that is reflected in velocities of Rayleigh waves can be enhanced by
metamorphic and deformational processes that produce fabrics such as schistosity and can form extensive regions of foliated
and laminated rocks (Okaya et al., 2018). Well documented example of a laminated lowermost crust provides the refraction
and wide-angle reflection experiment along profile ALP 01 (Brückl et al., 2007), parallel to the EASI profile (see Fig. 14c,
80-280 km). Several kilometre-thick laminated transition zone above the Moho is interpreted within the BM northward of
the contact with the Alps.




Assuming a mafic composition (amphibolite-facies metamorphic rocks), a ductile flow in the lower crust is probably
controlled by the deformation of plagioclase feldspar. Tendency of the lower crust to flow (e.g., Hacker et al., 2015) is
controlled by its rheology. The onset of plagioclase viscous behaviour corresponds to temperature of ~450°C (Cole et al.,
2007). Based on heat-flow measurements in western Bohemia (Šafanda et al., 2007), such temperature can be reached at
depths around 20 km. Of course, the present-day depth of the top of the ductile lower crust, which shallows up to ~15 km at
some places, was modified by denudation that is estimated at about 7 km in the western BM (de Wall et al., 2019). On the
other hand, the top of the lower crust deepens to ~28 km (Fig. 10) in the south-eastern part of the BM core, which probably
reflects thickening of the crust by Variscan (first oceanic, later continental) subductions (Medaris et al., 2006; Finger et al.,
2007). Consequently, a thermal gradient was lowered due to subductions of cold materials and by the time elapsed since
these events took place, and thus the onset of ductile flow might have been deeper in the geological past

Decoupling between the upper and lower crust due to rheological contrast between the two thermally different levels was
suggested in the French Pyrenees (Denele et al., 2009). Similar model of mechanical decoupling between the brittle upper
crust and a "hot deeper crustal level" is used to explain the crustal shortening during the India-Asia collision (Van Buer et
al., 2015). Many seismic and magneto-telluric experiments within Tibet provide data that require models characterised by a
less viscous mid-lower crust in-between more viscous upper crust and mantle lithosphere (e.g., Klemperer, 2006). Such
interpretations agree with the widely accepted "jelly sandwich" model of Burov and Watts (2006) that can explain best the
gross structural styles of plate tectonic collisional systems and also a large-scale flexure of the India Plate during the
collision (e.g., Hetényi et al., 2006).

**6.2.3 Regional distribution of VDI and late Variscan strike-slip tectonics**

Most of the BM crust and its southern and western surroundings are characterized by the abrupt shear velocity decrease in
the lower crust exceeding -0.4 km s$^{-1}$ at some places (see Fig. 10). Though the amplitude of the velocity drop and depth of
the drop can be clustered according to individual tectonic units, there is only a weak relation between the depth of the VDI or
the thickness of the LPC layer and the crustal thickness. The shallowest VDI (green in Fig. 10) is mostly mapped in regions
with thinner crust (Fig. 8). On the other hand, the most significant velocity drop appears both within regions with the thin
and thick crust.

The VDI horizon, marked at sites where the velocity difference attains at least -0.1 km s$^{-1}$, is interrupted at several zones,
where the negative $v_S$ drop does not reach such value or is absent. Evidently, such zones appear also as regions with no
difference between the upper and lower crust velocities (Fig. 10) and often with a local velocity increase in the lowermost
crust.





Figure 15 depicts a NW-SE cross section through the core of the BM, showing the $v_S$ in the crust of the four major tectonic units, former microplates assembled during the Variscan orogeny. The horizon of the VDI is interrupted above mantle sutures and near the boundaries of crustal units, often laterally shifted from their mantle counterparts. The sutures probably served as subduction channels that later became paths, along which subducted rocks were brought to the upper crust and

provided magma for large granitoid massifs like the Central Bohemian Plutonic Complex or the Moldanubian Batholith (Babuška and Plomerová, 2013). The interruption of the VDI and relatively increased velocities in the lower crust correlate with the deep sutures also on other cross sections (Fig. 14b-c). No VDI interruption within the BM occurs along the S04 (Fig.14a) which runs parallel to the EFZ and does not cross any distinct suture. This feature supports the importance of the late-Variscan strike-slip tectonics, in which the Elbe shear system played important role (Elter et al., 2020).


Contraction of the eastern Variscides caused by the north-westward advance of Gondwana resulted in a collisional shortening of the BM (Franke et al., 2017). The contraction was compensated by Late-Variscan strike-slip zones that considerably modified an original assembly of the BM (Pitra et al., 1999). Field measurements by these authors document dextral strike-slip movements along the CBSZ. This is in agreement with the model of the westward movement of a part of

the TB lithosphere along the WSW oriented sutures on both sides of the TB unit (Babuška and Plomerová, 2017).

The model we present in Fig. 16 can explain the spatial distribution of VDI, why it is interrupted at places where the velocity decrease is weaker than -0.1 km s$^{-1}$ or is absent. Most of the interruptions correlate with the deep boundaries of the tectonic units, e.g., ST/TB, TB/MD, and MSD/BV (Babuška and Plomerová, 2013). Late Variscan strike-slip movements preferably

took place along such boundaries and most likely modified the sub-horizontal fabric of the lower crust by a system of parallel, deep-reaching faults. Strike-slip zones, like the CBSZ or the suture between the ST and TB units (see Fig. 15), undoubtedly cut the whole crust by sub-vertical faults that continue towards the mantle lithosphere. Movements along these zones disrupted the horizontal fabric and the VTI in the lower crust. Fabrics around the block boundaries might be modified locally either into a more complex symmetry, e.g., the orthorhombic symmetry with "fast" horizontal axis oriented in

direction of movement, or might be entirely overwritten.

## 7 Conclusions

We present the first high-resolution 3-D shear velocity model of the BM crust and a map of Moho depths retrieved from ambient noise tomography. The model is based on recordings of more than 400 stations operated in several passive seismic experiments organised during the last two decades, including the large-scale AlpArray network (Hetényi et al., 2018a). The

shear velocities modelled from dispersion curves of Rayleigh waves are ~0.2 km s$^{-1}$ higher in the BM upper crust than in its surroundings with the highest velocities in the Moldanubian part. The region north of the Elbe Fault Zone exhibits relatively lower velocities. The presented 3-D model provides compelling evidence of a regional-scale velocity distribution.

The Moho in the ambient noise tomography appears in the whole region as a sharp interface, across which the lower-crust

velocities (~3.5-3.7 km s$^{-1}$) attain the upper mantle velocities.(~4.2 km s$^{-1}$). The Moho shallows up to ~28 km beneath the Eger Rift and its crossing with the Elbe Fault Zone. The Moho deepens down to ~40 km in the Moldanubian part of the BM. In general, the Cadomian part of the region has a thinner crust, while the crust assembled, or tectonically transformed in the Variscan period, is thicker. The Moho discontinuity, as the dynamic phenomenon, retains remnants of Variscan subductions imprinted in bands of local crustal thickenings.


The dominant feature of the presented 3-D velocity model is relatively low $v_S$ in the lower crust that is in contradiction with the high $v_P$ modelled in the BM by the Control Source Seismic profiles. We explain this seeming disagreement by anisotropic structure of the lower crust characterized by a transversely isotropic medium with vertical 'low-velocity' symmetry axis. We estimate strength of anisotropy at 3–9 % in dependence on the average $v_S$. Considering the observed

anisotropy and both $v_P$ and $v_S$ interpreted from different field seismic experiments, amphibolite is the best candidate for the composition of the BM lower crust.

Thanks to the high density of data, thorough processing, enhanced depth sensitivity of trans-dimensional tomographic inversion by layer stripping, layer averaging and allowing for a velocity decrease with depth during the stochastic inversion,

we were able to retrieve, for the first time, a distinct intra-crustal interface (VDI) with a velocity decrease as strong as to -0.4 km s$^{-1}$ in the lower part of the crust at depths 18–30 km. Particularly the averaging of the best-fitting models over layers instead of the commonly used averaging over fixed depth, proved to be crucial for the detection of velocity interfaces.

The VDI is interrupted around boundaries of the crustal units, usually above locally increased velocities in the lowermost

crust. Boundaries of the tectonic units likely served as subduction channels and partly as strike-slip zones in late-Variscan tectonic processes. Such zones can represent channels through which the high-velocity mantle material intruded into the lower crust. Portions of subducted and later molten crustal rocks have percolated upwards providing magma to subsequently form granitoid massifs like the Central Bohemian Plutonic Complex and the Moldanubian Batholith. Moreover, the late Variscan strike-slip movements rejuvenated such boundaries. The movements might modify or erase the fabric of the lower

crust and locally also interrupt the VDI horizon. This horizon and the lower-crust anisotropy probably reflect a rheological contrast between the formerly more viscous upper crust and less viscous lower crust.

The presented 3-D shear velocity model of the BM crust is homogeneous in the whole massif with unprecedented resolution. It tracks the variations of the velocities and the Moho depth at a grid of 0.3° E-W by 0.2° N-S. Understanding anisotropy in

the lower crust can be further improved by joined inversion of Rayleigh and Love waves, along with the receiver functions.



The presented model is potentially of a great use in other geophysical studies and modern geological, petrological and geochemical interpretations.

**Team list**. The complete member list of the AlpArray Working Group can be found at http://www.alparray.ethz.ch.

**Acknowledgements.** Research within this study was supported by the grant of the Czech Academy of Sciences M100121201 and projects CzechGeo/EPOS-Sci CZ.02.1.01/0.0/0.0/16_013/0001800 (OP RDE), CzechGeo/EPOS LM2010008 and LM2015079, and MEYS-COST CZ LD15029. We acknowledge the operation of the temporary seismic network XT of the AlpArray-EASI complementary experiment (Hetényi et al., 2018a) and the AlpArray Seismic Network Z3 (2015) as well several previous regional experiments in the Bohemian Massif. The authors would like to thank to the AlpArray Seismic Network Team: György HETÉNYI, Rafael ABREU, Ivo ALLEGRETTI, Maria-Theresia APOLONER, Coralie AUBERT, Simon BESANÇON, Maxime BÈS DE BERC, Götz BOKELMANN, Didier BRUNEL, Marco CAPELLO, Martina ČARMAN, Adriano CAVALIERE, Jérôme CHÈZE, Claudio CHIARABBA, John CLINTON, Glenn COUGOULAT, Wayne C. CRAWFORD, Luigia CRISTIANO, Tibor CZIFRA, Ezio D'ALEMA, Stefania DANESI, Romuald DANIEL, Anke DANNOWSKI, Iva DASOVIĆ, Anne DESCHAMPS, Jean-Xavier DESSA, Cécile DOUBRE, Sven EGDORF, ETHZ-SED Electronics Lab, Tomislav FIKET, Kasper FISCHER, Wolfgang FRIEDERICH, Florian FUCHS, Sigward FUNKE, Domenico GIARDINI, Aladino GOVONI, Zoltán GRÁCZER, Gidera GRÖSCHL, Stefan HEIMERS, Ben HEIT, Davorka HERAK, Marijan HERAK, Johann HUBER, Dejan JARIĆ, Petr JEDLIČKA, Yan JIA, Hélène JUND, Edi KISSLING, Stefan KLINGEN, Bernhard KLOTZ, Petr KOLÍNSKÝ, Heidrun KOPP, Michael KORN, Josef KOTEK, Lothar KÜHNE, Krešo KUK, Dietrich LANGE, Jürgen LOOS, Sara LOVATI, Deny MALENGROS, Lucia MARGHERITI, Christophe MARON, Xavier MARTIN, Marco MASSA, Francesco MAZZARINI, Thomas MEIER, Laurent MÉTRAL, Irene MOLINARI, Milena MORETTI, Anna NARDI, Jurij PAHOR, Anne PAUL, Catherine PÉQUEGNAT, Daniel PETERSEN, Damiano PESARESI, Davide PICCININI, Claudia PIROMALLO, Thomas PLENEFISCH, Jaroslava PLOMEROVÁ, Silvia PONDRELLI, Snježan PREVOLNIK, Roman RACINE, Marc RÉGNIER, Miriam REISS, Joachim RITTER, Georg RÜMPKER, Simone SALIMBENI, Marco SANTULIN, Werner SCHERER, Sven SCHIPPKUS, Detlef SCHULTE-KORTNACK, Vesna ŠIPKA, Stefano SOLARINO, Daniele SPALLAROSSA, Kathrin SPIEKER, Josip STIPČEVIĆ, Angelo STROLLO, Bálint SÜLE, Gyöngyvér SZANYI, Eszter SZŰCS, Christine THOMAS, Martin THORWART, Frederik TILMANN, Stefan UEDING, Massimiliano VALLOCCHIA, Luděk VECSEY, René VOIGT, Joachim WASSERMANN, Zoltán WÉBER, Christian WEIDLE, Viktor WESZTERGOM, Gauthier WEYLAND, Stefan WIEMER, Felix WOLF, David WOLYNIEC, Thomas ZIEKE, Mladen ŽIVČIĆ and Helena ŽLEBČÍKOVÁ. Also, we would like to thank all the network operators providing data to the EIDA archive (http://www.orfeus-eu.org/eida). The authors are grateful to makers of software packages which we used for data processing and creating of graphics: Obspy (Beyreuther et al., 2010), MSNoise (Lecocq et al., 2014), FMST (Rawlinson, 2005), Geopsy (Wathelet et al., 2020), Python Matplotlib (Hunter, 2007), and GMT (Wessel and Smith, 1991).





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

**Table 1. Range of parameters for Neighbourhood Algorithm (NA) stochastic inversion.**

| Layer | bottom depth (km) | $v_S$ (km s$^{-1}$) | $v_P$ (km s$^{-1}$) | density (g cm$^{-3}$) | velocity gradient | number of sub-layers | layer stripping |
|---|---|---|---|---|---|---|---|
| near surface 1 | 0.1–2 | 1.2–2.5  3.0–6.5 | 1.5–3.5 | constant velocity | 1 | 0 | – – – |
| near surface 2 | 1–5 | 1.5–3.0  3.0–6.5 | 1.5–3.5 | constant velocity | 1 | 0 | – – – |
| near surface 3 | 2–18 | 2.0–3.5  4.0–6.5 | 2.0–3.5 | linear gradient | 5 | 0 | 1 – – |
| upper part of the crust | 13–30 | 3.0–4.0  5.5–7.5 | 2.0–4.0 | linear gradient | 7 | 0 | 1 2 – |
| lower part of the crust | 21–45 | 3.0–4.0  5.5–7.5 | 2.0–4.0 | linear gradient | 5 | 0 | 1 2 3 |
| uppermost mantle | 52–80 | 4.0–4.7  7.0–9.0 | 2.5–4.5 | linear gradient | 5 | 0 | 1 2 3 |
| half space | – | 4.0–5.0  7.0–9.0 | 2.5–4.5 | constant velocity | 1 | 0 | 1 2 3 |

**Table 2. Models of the crust beneath the KHC station (Moldanubian Unit)**

| | Spectral ratio method Červený et al., 1977 | | | 3D ANT model 49.13N 13.58E | |
|---|---|---|---|---|---|
| | $v_P$ | $v_S$ | H(km) | $v_S$ | H(km) |
| | (km/s) | | | (km/s) | |
| Upper crust | 6.0 | 3.5 | 20 | 3.5 | 25 |
| Lower crust | 6.8 | 3.9 | 35 | 3.45 | 34 |
| Half-space | 7.8-8.2 | 4.5-4.7 | - | 4.2 | - |

**Figure 1: Simplified overview of tectonic units of the Bohemian Massif (BM) and fault systems. Modelled region is framed in blue in the inset. Red frame denotes area of seismic stations used in the study.**

**Figure 2: Map of the AlpArray seismic stations (red), temporary stations from previous passive experiments (orange) and permanent stations available in the European Integrated Data Archive (http://www.orfeus-eu.org/eida) in blue. Modelled region is framed, contours of geological units from Figure 1 are shown.**

**Figure 3: Examples of cross-correlation functions (CCF) between data from station CZ.PRU and all permanent stations for summer (a) and winter (b) periods. The blue area marks window used to measure RMS amplitude of coherent signal and the red area marks window used to measure RMS amplitude of non-coherent noise. Azimuth-interstation distance rose diagrams present signal-to-noise ratio (c) and source directivity calculated as ratios of the causal and acausal parts of the CCF signals (d) for all permanent stations in the region for winter and summer period (left and right diagrams, respectively).**

**Figure 4: Flow chart of the method and data processing.**

**Figure 5: Example of dispersion curve for the pair of stations Z3.A001A–Z3.A354A from frequency-time analysis (FTAN) (a) and Rayleigh wave group velocities at 19 s period between all station pairs (b).**

**Figure 6: Dispersion curves (right) computed at 14.4° E 50.7° N for velocity models (left) averaged either over fixed depths (blue) or over layer indexes (red), without (a) or with (b,c) layer successive stripping and with velocity decrease allowed (a,b) or not allowed (c) in the inversion. Tests of gradational vs. sharp Moho (d) limit thickness of the potential transitional layer to 2 km. The**





yellow to green field marks 10 % of the best models from 400 000 computed ones, 1 % of the best models is framed in the thin black line. The first quartile, median and the third quartile display of misfits between the input dispersion curves and those computed for all final 1-D models (see part (b), red curve) after each step of layer stripping (e) in two period ranges 6–48 s and 12–37 s (blue and orange, respectively). The misfit improves linearly with each run of layer stripping.

Figure 7: Examples of shear velocity slices through the final model at 10, 25 and 32 km depths.

Figure 8: Thickness of the BM crust from ambient noise tomography (a). The Moho depth 38 km is contoured. Average velocities in the upper part of the crust (UPC) (b) and the lower part of the crust (LPC) (c).

Figure 9: Examples of vS models (a), characteristic for individual tectonic units (for the abbreviations see Fig. 1 and the text) and corresponding dispersion curves (b). Blue curve is the average model calculated from the 10 % of the best fitting models from the 400 000 models in total (yellow-to-green curves)and their corresponding dispersion curves (b). Thin black curves frame 1 % of the best models. Dispersion curve input into the inversion is in grey dots (b).

Figure 10: Depth of velocity drop interface (VDI) along with differences in velocity gradients (coloured background) in the upper
and lower part of the crust (UPC, LPC). Size of the circles is proportional to the amplitude of the velocity drop across the VDI. Note the deeper VDI in the Moldanubian and Brunovistulian units with the thick crust.

Figure 11: 3-D view of the $v_S$ ANT model of the BM from south-west. The yellow segments denote the velocity drop interface (VDI) at sites with velocity decrease at least -0.1 km/s. The Moho depth estimated from the Ps–P delay times in receiver functions calculated in this study for all the stations is shown in red. Profile locations and names are shown in the inset. For more views see
S1–S4.

Figure 12: Checkerboard tests (b) with cell sizes of 0.6° E-W by 0.4° N-S, 0.9° E-W by 0.6° N-S and 1.2° E-W by 0.8° N-S (a) for ray path coverages at periods 6 s, 19 s, and 48 s. Hit-counts (c) – number of rays crossing each cell centred around 1–D inversion grid of 0.3° E-W by 0.2° N-S.

Figure 13: Histogram of the ray path coverage in dependence on period (a). Average sensitivity kernels (b) calculated for velocity
profile at all cells in the BM. Black line in green box marks depth of maximum sensitivity and size of the box marks zone of more than 50 % of maximum sensitivity in each period. Orange curve shows the RMS of travel time residuals after the fast marching inversion (FMST) for all station pairs.

Figure 14: Cross sections through the 3–D $v_S$ ANT model (coloured background) along five CSS profiles. White lines mark velocity boundaries according to CSS models ALP01 (Brückl et al., 2007), CEL09 (Hrubcová et al., 2005), CEL10 (Hrubcová and Środa,
2008), S01 (Grad et al., 2008) and S04 (Hrubcová and Środa, 2010) (see inset for locations). The Moho depth, estimated from the Ps–P delay times in receiver functions calculated in this study for all the stations, is in red. Magenta line in part (e) is the Moho along the AlpArray–EASI by Hetényi et al. (2018b).

Figure 15: NW–SE oriented cross section through the core of the BM (see Fig. 12 for location of the cross section), across the NE–SW trend of the Variscan structures. Note the places of increased vS in the lower crust beneath the VDI interruptions. They can
mark pathways (arrows) developed along the deep Variscan suture zones and boundaries of the individual tectonic units of the crust. The pathways served for magma to percolate and feed the granitoid massifs (Central Bohemian Plutonic Complex and Moldanubian Batholith). The Moho depth estimated from the Ps–P delay times in receiver functions calculated in this study for all the stations is in red.

Figure 16: Scheme of tectonic processes affecting the lower-crust fabric. (a) Bottom-driven NW–SE shortening of the core of the
BM produced or enhanced already existing sub horizontal foliation (fabric) within the lower crust rocks. (b) Late-Variscan strike-slip fault systems often developed along the pre-existing deep sutures cutting the whole lithosphere. Movement along the sutures might locally modify, overprint or erase the sub-horizontal fabric of the BM lower crust.



**Table 1. Range of parameters for Neighbourhood Algorithm (NA) stochastic inversion.**

| Layer | bottom depth (km) | $v_S$ (km s$^{-1}$) | $v_P$ (km s$^{-1}$) | density (g cm$^{-3}$) | velocity gradient | number of sub-layers | layer stripping |
|---|---|---|---|---|---|---|---|
| near surface 1 | 0.1–2 | 1.2–2.5 | 3.0–6.5 | 1.5–3.5 | constant velocity | 1 | 0 – – – |
| near surface 2 | 1–5 | 1.5–3.0 | 3.0–6.5 | 1.5–3.5 | constant velocity | 1 | 0 – – – |
| near surface 3 | 2–18 | 2.0–3.5 | 4.0–6.5 | 2.0–3.5 | linear gradient | 5 | 0 1 – – |
| upper part of the crust | 13–30 | 3.0–4.0 | 5.5–7.5 | 2.0–4.0 | linear gradient | 7 | 0 1 2 – |
| lower part of the crust | 21–45 | 3.0–4.0 | 5.5–7.5 | 2.0–4.0 | linear gradient | 5 | 0 1 2 3 |
| uppermost mantle | 52–80 | 4.0–4.7 | 7.0–9.0 | 2.5–4.5 | linear gradient | 5 | 0 1 2 3 |
| half space | – | 4.0–5.0 | 7.0–9.0 | 2.5–4.5 | constant velocity | 1 | 0 1 2 3 |


**Table 2. Models of the crust beneath the KHC station (Moldanubian Unit)**

| | Spectral ratio method Červený et al., 1977 | | | 3D ANT model 49.13N 13.58E | |
|---|---|---|---|---|---|
| | $v_P$ | $v_S$ | H(km) | $v_S$ | H(km) |
| | (km/s) | | | (km/s) | |
| Upper crust | 6.0 | 3.5 | 20 | 3.5 | 25 |
| Lower crust | 6.8 | 3.9 | 35 | 3.45 | 34 |
| Half-space | 7.8-8.2 | 4.5-4.7 | - | 4.2 | - |




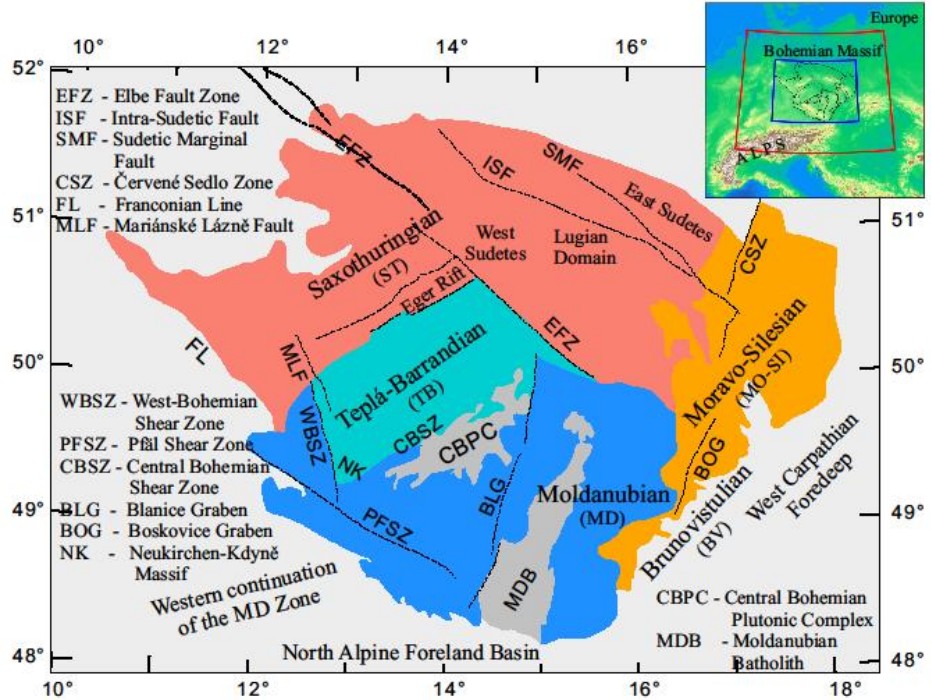

**Figure 1: Simplified overview of tectonic units of the Bohemian Massif (BM) and fault systems. Modelled region is framed in blue in the inset. Red frame denotes area of seismic stations used in the study.**





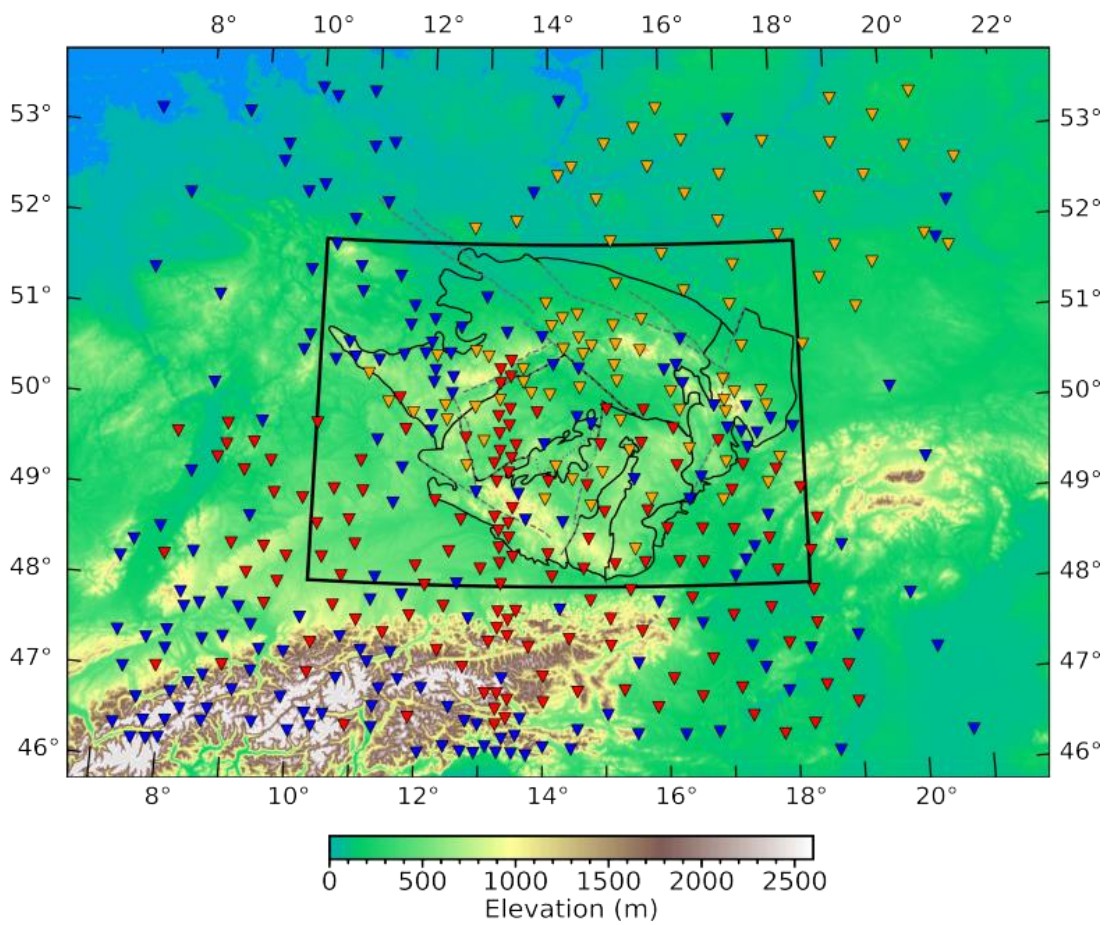

**Figure 2: Map of the AlpArray seismic stations (red), temporary stations from previous passive experiments (orange) and permanent stations available in the European Integrated Data Archive (http://www.orfeus-eu.org/eida) in blue. Modelled region is framed, contours of geological units from Figure 1 are shown.**





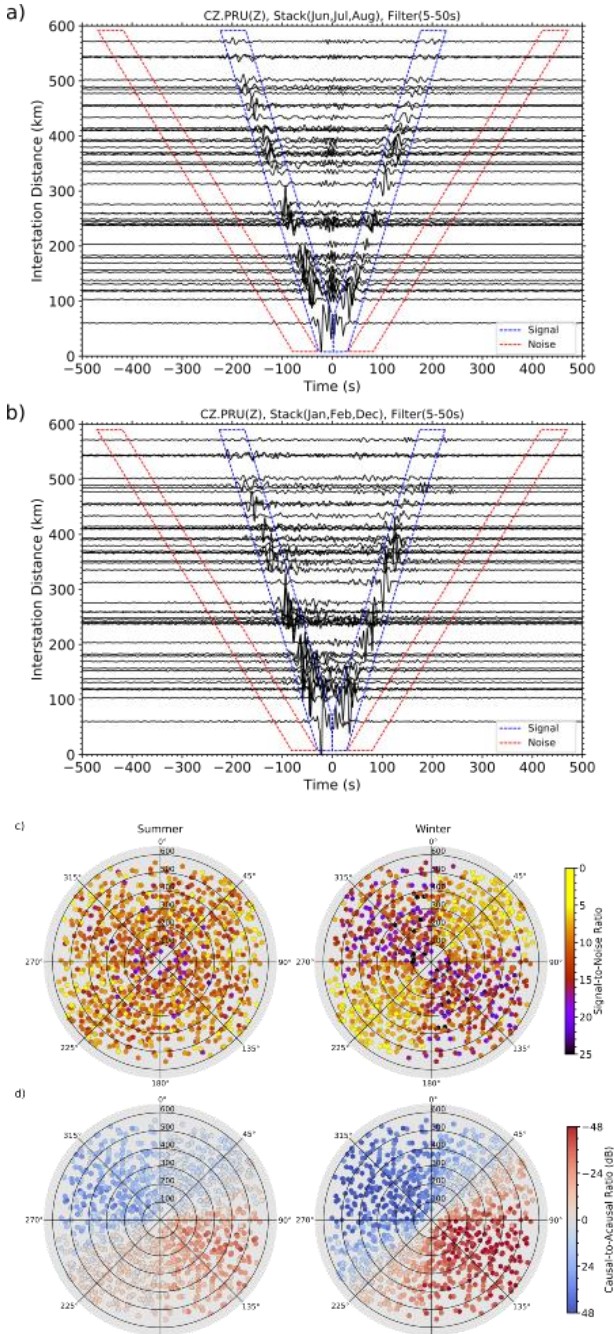

**Figure 3: Examples of cross-correlation functions (CCF) between data from station CZ.PRU and all permanent stations for summer (a) and winter (b) periods. The blue area marks window used to measure RMS amplitude of coherent signal and the red area marks window used to measure RMS amplitude of non-coherent noise. Azimuth-interstation distance rose diagrams present signal-to-noise ratio (c) and source directivity calculated as ratios of the causal and acausal parts of the CCF signals (d) for all permanent stations in the region for winter and summer period (left and right diagrams, respectively).**



**Figure 4: Flow chart of the method and data processing.**


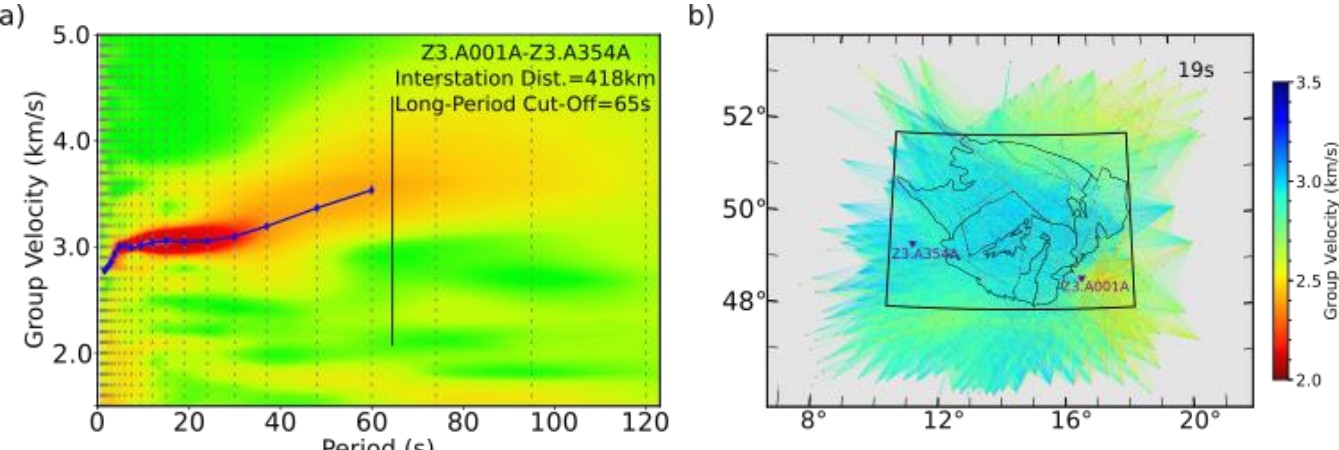

**Figure 5: Example of dispersion curve for the pair of stations Z3.A001A–Z3.A354A from frequency-time analysis (FTAN) (a) and Rayleigh wave group velocities at 19 s period between all station pairs (b).**





**Figure 6:** Dispersion curves (right) computed at 14.4° E 50.7° N for velocity models (left) averaged either over fixed depths (blue) or over layer indexes (red), without (a) or with (b,c) layer successive stripping and with velocity decrease allowed (a,b) or not allowed (c) in the inversion. Tests of gradational vs. sharp Moho (d) limit thickness of the potential transitional layer to 2 km. The yellow to green field marks 10 % of the best models from 400 000 computed ones, 1 % of the best models is framed in the thin black line. The first quartile, median and the third quartile display of misfits between the input dispersion curves and those computed for all final 1-D models (see part (b), red curve) after each step of layer stripping (e) in two period ranges 6–48 s and 12–37 s (blue and orange, respectively). The misfit improves linearly with each run of layer stripping.



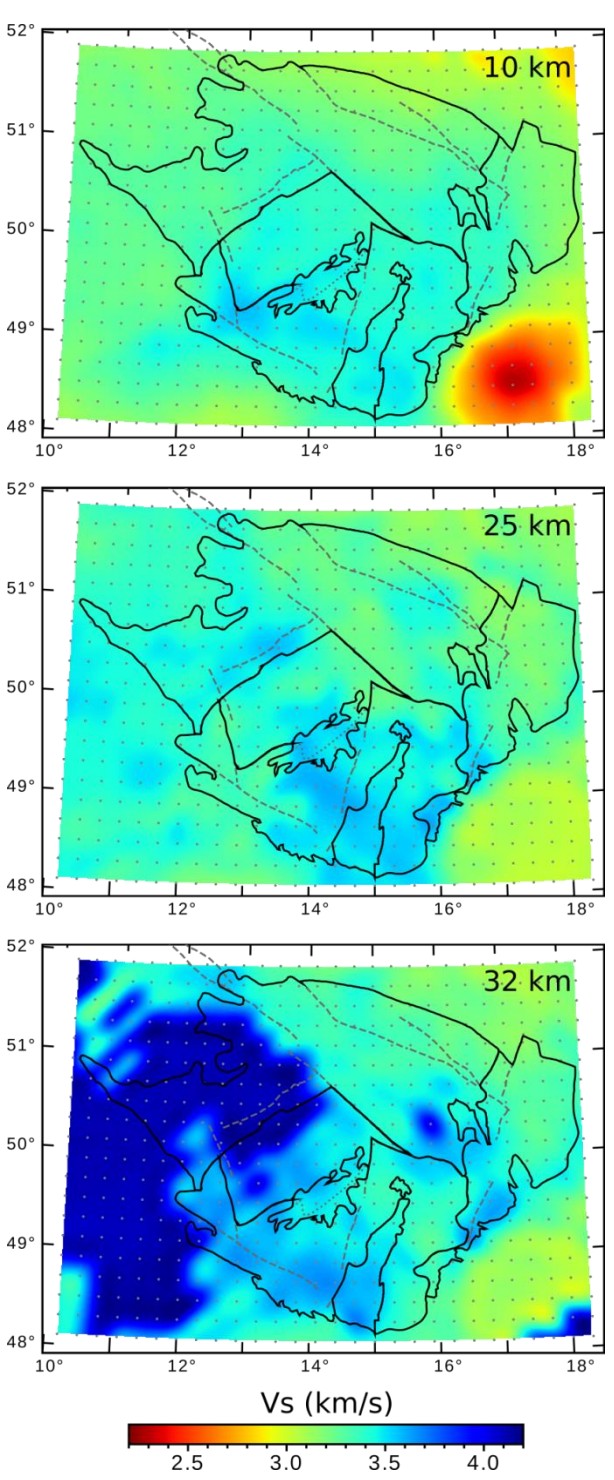

**Figure 7: Examples of shear velocity slices through the final model at 10, 25 and 32 km depths.**




**Figure 8: Thickness of the BM crust from ambient noise tomography (a). The Moho depth 38 km is contoured. Average velocities in the upper part of the crust (UPC) (b) and the lower part of the crust (LPC) (c).**





a)




b)

**Figure 9: Examples of vS models (a), characteristic for individual tectonic units (for the abbreviations see Fig. 1 and the text) and corresponding dispersion curves (b). Blue curve is the average model calculated from the 10 % of the best fitting models from the**
**400 000 models in total (yellow-to-green curves)and their corresponding dispersion curves (b). Thin black curves frame 1 % of the best models. Dispersion curve input into the inversion is in grey dots (b).**





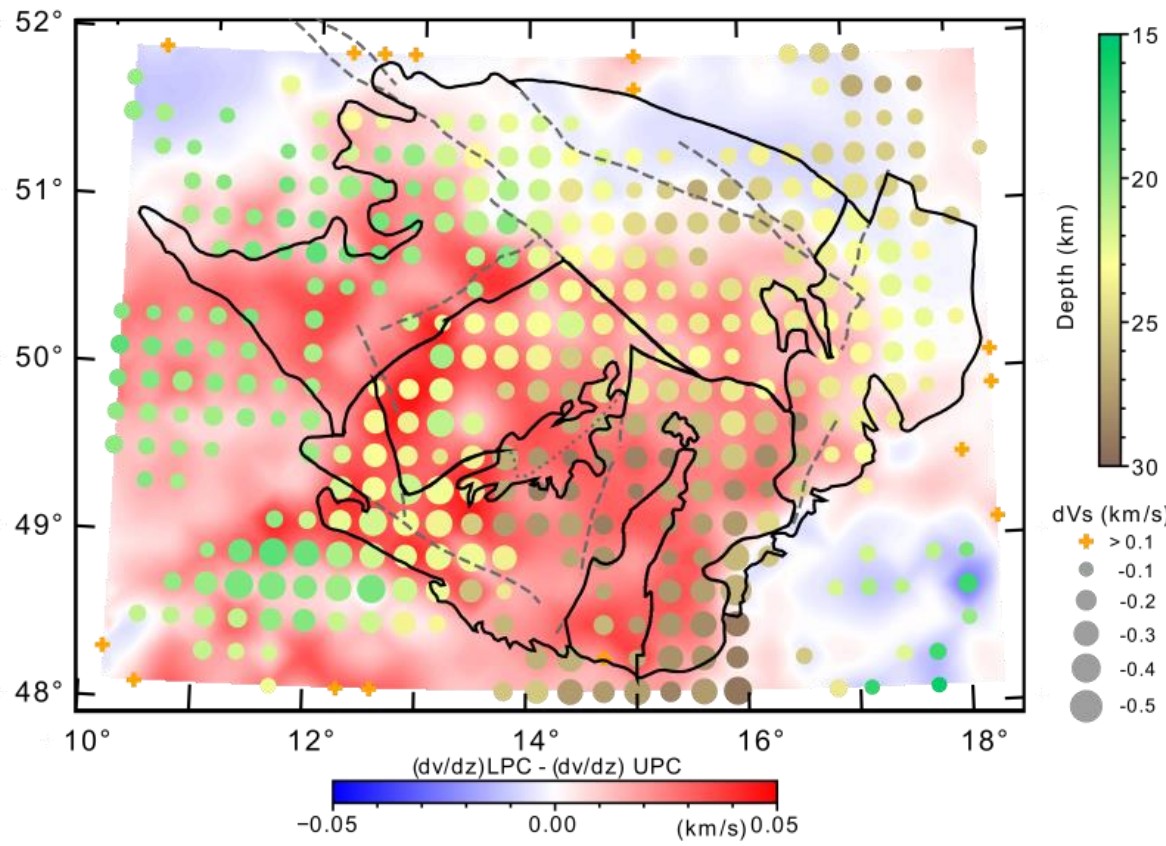

**Figure 10: Depth of velocity drop interface (VDI) along with differences in velocity gradients (coloured background) in the upper and lower part of the crust (UPC, LPC). Size of the circles is proportional to the amplitude of the velocity drop across the VDI. Note the deeper VDI in the Moldanubian and Brunovistulian units with the thick crust.**





**Figure 11: 3-D view of the $v_S$ ANT model of the BM from south-west. The yellow segments denote the velocity drop interface (VDI) at sites with velocity decrease at least -0.1 km/s. The Moho depth estimated from the Ps–P delay times in receiver functions calculated in this study for all the stations is shown in red. Profile locations and names are shown in the inset. For more views see S1–S4.**





**Figure 12: Checkerboard tests (b) with cell sizes of  0.6° E-W by 0.4° N-S,  0.9° E-W by 0.6° N-S and  1.2° E-W by 0.8° N-S (a) for ray path coverages at periods 6 s, 19 s, and 48 s. Hit-counts (c) – number of rays crossing each cell centred around 1–D inversion grid of 0.3° E-W by 0.2° N-S.**




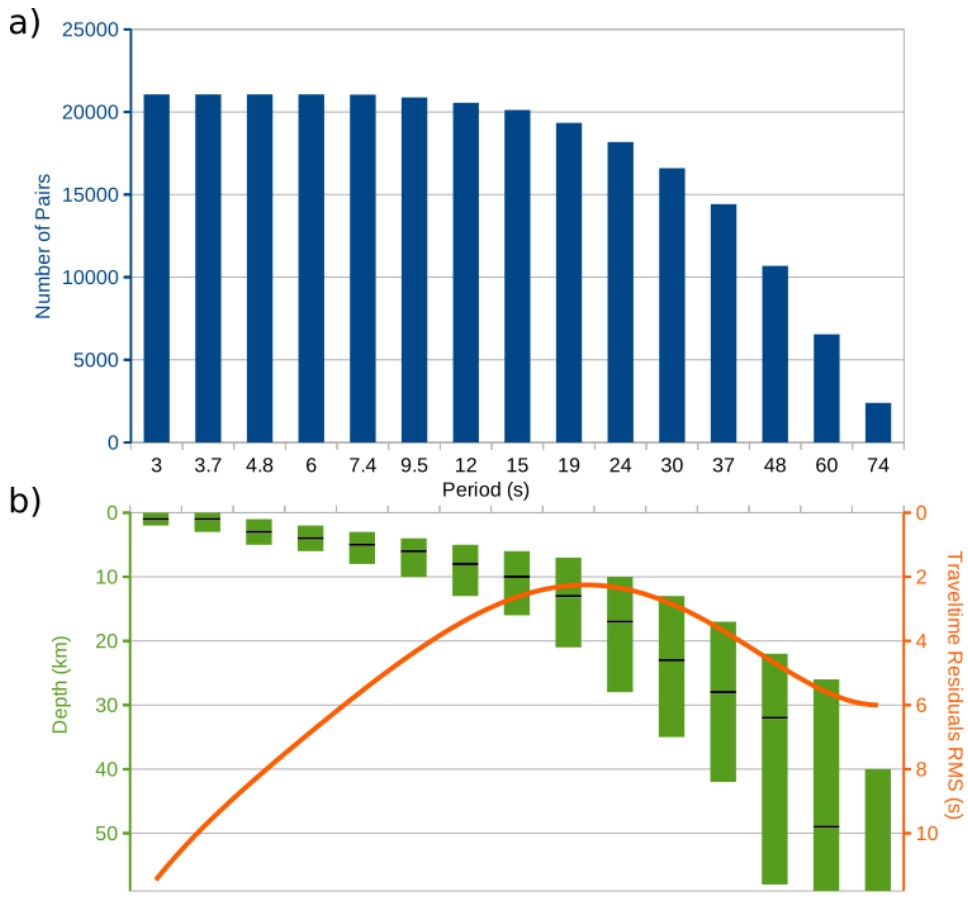


**Figure 13: Histogram of the ray path coverage in dependence on period (a). Average sensitivity kernels (b) calculated for velocity profile at all cells in the BM. Black line in green box marks depth of maximum sensitivity and size of the box marks zone of more than 50 % of maximum sensitivity in each period. Orange curve shows the RMS of travel time residuals after the fast marching inversion (FMST) for all station pairs.**






**Figure 14: Cross sections through the 3–D v$_S$ ANT model (coloured background) along five CSS profiles. White lines mark velocity boundaries according to CSS models ALP01 (Brückl et al., 2007), CEL09 (Hrubcová et al., 2005), CEL10 (Hrubcová and Środa, 2008), S01 (Grad et al., 2008) and S04 (Hrubcová and Środa, 2010) (see inset for locations). The Moho depth, estimated from the Ps–P delay times in receiver functions calculated in this study for all the stations, is in red. Magenta line in part (e) is the Moho along the AlpArray–EASI by Hetényi et al. (2018b).**



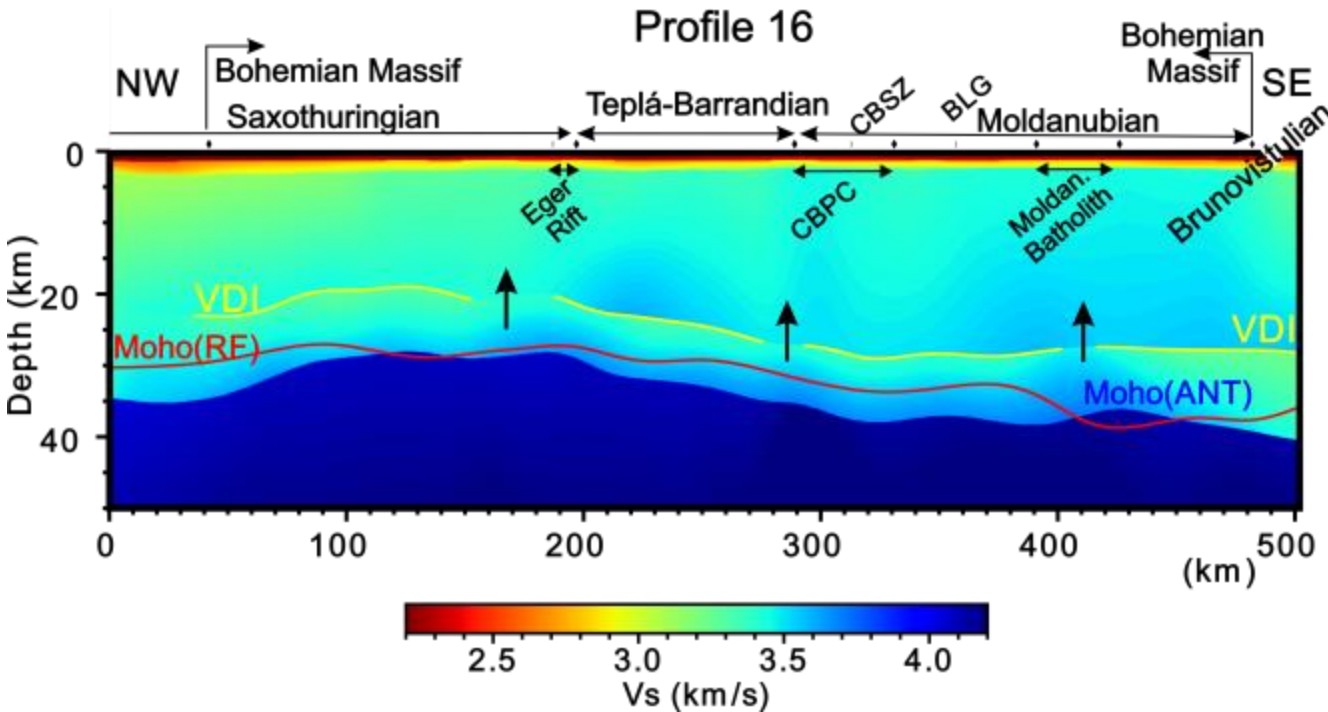


**Figure 15: NW–SE oriented cross section through the core of the BM (see Fig. 12 for location of the cross section), across the NE–SW trend of the Variscan structures. Note the places of increased vS in the lower crust beneath the VDI interruptions. They can mark pathways (arrows) developed along the deep Variscan suture zones and boundaries of the individual tectonic units of the crust. The pathways served for magma to percolate and feed the granitoid massifs (Central Bohemian Plutonic Complex and**
**Moldanubian Batholith). The Moho depth estimated from the Ps–P delay times in receiver functions calculated in this study for all the stations is in red.**



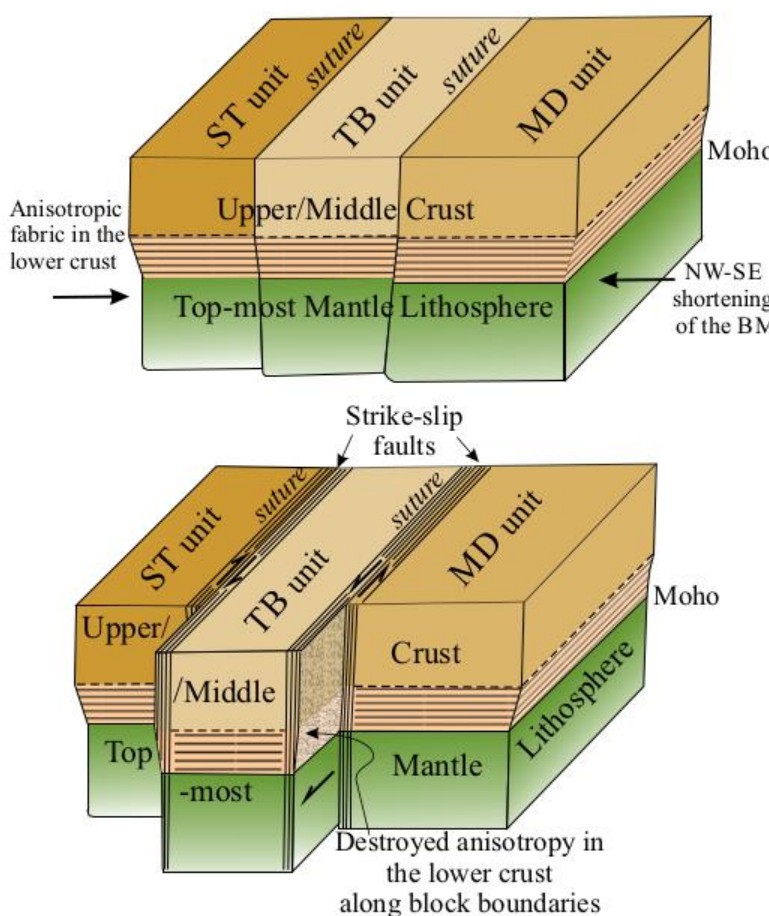

**Figure 16: Scheme of tectonic processes affecting the lower-crust fabric. (a) Bottom-driven NW–SE shortening of the core of the BM produced or enhanced already existing sub horizontal foliation (fabric) within the lower crust rocks. (b) Late-Variscan strike-slip fault systems often developed along the pre-existing deep sutures cutting the whole lithosphere. Movement along the sutures might locally modify, overprint or erase the sub-horizontal fabric of the BM lower crust.**