# Peer review of "Transversely Isotropic Lower Crust of Variscan Central Europe imaged by Ambient Noise Tomography of the Bohemian Massif"

_Solid Earth, 2020_

## Referee Comment (RC1) · Mariusz Majdanski (Referee) · 29 Nov 2020

I am a physicist with experience in modelling of the crustal structure using a variety of seismic methods, but mostly specialised in P-wave controlled source data, and the uncertainty analysis of the structure models.

General Comments

This is a very well written manuscript describing a new high-resolution 3D model of the shear wave velocities in the Bohemian Massif, based on ambient noise data. It contains an informative introduction, followed by short information about used data

(selected 24 258 station pairs for ambient noise processing) and detailed description of used methods (an iterative fast marching method resulting in shear velocity maps and a stochastic inversion of dispersion curves for a collection of multi-layered shear velocity models). Furter it describe the results with a large number of figures and useful supplements, resolution and sensitivity tests. The main content of the article is the discussion, that itself is a review of all important articles describing the structure of the Bohemian Massif and especially its lower crust.

The manuscript is long, but clearly describe data analysis, chosen assumptions and methods limitations. It is interesting as it presents the low shear wave velocity anomaly in the lower crust of the whole Bohemian Massif in contradiction to everything that was previously published, and explain that by anisotropy of seismic velocity. As an easy improvement, this manuscript could be shortened and even some figures (e.g. fig.5, 7, 8, 13) could be removed without harming the main message.

Specific comments

1. Reduction of shear wave velocities in the lower crust is a new and surprising result. No such effect was visible in previous P-wave velocity results from CSS experiments. It is, of course, possible that P and S waves have different characteristics, but would it be possible to verify S-wave velocities in the lower crust using existing CSS data? It would be less precise than P-waves, but still, such characteristics should be clearly visible.

The most surprising is the existence of those lower velocities in the whole area. As shown using P-wave CSS at profiles CEL09 (Hrubcova et al. 2005 JGR doi: 10.1029/2004JB003080), and S02, S03 (Majdanski et al. 2006 Tectonophysics doi: 10.1016/j.tecto.2005.10.042) anomalous P-wave velocities in form of layered lower crust were recognized in part of profiles, but not globally. Those areas were interpreter as remelting of the lower crust or magmatic underplating (Majdański et al., Tectono-physics, doi: 10.1016/j.tecto.2007.02.015). Is it really a global feature of Vs of BM?

2. Despite the possible difference in P and S-waves velocities in the lower crust that are possible the boundary of the lower crust, both upper and lower (Moho) should match the P-wave models, at least in range of the uncertainty of used methods. As shown they are quite different. The argument of 3D modelling advantage over 2D CSS is valid, but this effect is not that strong, because in the Bohemian Massif horizontal variations are not that significant. The question is, what level of S-wave anisotropy in the upper and middle crust would be needed to match P-wave models boundaries? As shown by Sroda 2006 GRL doi:10.1029/2006GL027701, for P-wave 10% anisotropy was observed.

3. I strongly disagree that the results of CSS should be neglected as less precise comparing to ambient noise methods. From personal experience, I am convinced that controlled source seismic and P-waves analysis is the most precise method of studying the crustal structure. The second one is CSS with S-waves (as they are less precise to recognize because arrive later, are converted and mixed). The next in precision are receiver function methods, and the least precise the Surface waves (dispersion analysis) methods. Authors refer to my paper (Majdański and Polkowski, 2014 PAGEOPH doi 10.1007/s00024-014-0840-9) as weakness of CSS. This paper shows the limitation of the layer-stripping approach. With previous paper (Majdański Geophysics doi: 10.1190/GEO2012-0280.1) it proves that the uncertainty of layer-stripping approach grows as dv*sqrt(n), where n is the number of stripping iterations. The same is valid for presented layer-stripping ANT in the discussed paper. So what is the final uncertainty of the presented model?

Small technical corrections:

The manuscript is partially written in the first person: "We picked", "We interpolate". Should be in third person.

28: remove space

166: remove space

405: remove . .

I hope that my comments will help to improve this manuscript.

With best regards,

Mariusz Majdański

---

## Referee Comment (RC2) · Amr El-Sharkawy (Referee) · 7 Dec 2020

General Comments

The authors conducted an ambient noise tomographic study for imaging the crust of the Bohemian Massif, central Europe. A new high resolution 3-D shear-wave crustal structure, constrained by about 21,066 Rayleigh wave group velocity dispersion curves in the period range from 2 to 74 s, is presented. In addition, estimates of the Moho topography across the region are calculated from surface wave dispersion measurements as well as Receiver Functions. Waveform data in the time period from 2002 to 2016, recorded by permanent and temporarily deployed seismic stations in the area

(e.g. Alp Array network, EASI network, etc.) are used in this study. The manuscript starts with a tectonic summary of Bohemian Massif and the previous studies. The authors then provide a brief description of the ambient noise cross correlation processing and the group velocity dispersion measurements that are followed by a two-step inversion approach for calculating the 3-D Vs model via period-dependent group velocity maps. The manuscript ends up with a detailed discussion of the outcomes that are furtherly compared with the previously available studies, mainly controlled source seismic profiling (CSS), in the area.

The manuscript is generally in a good shape but can largely be improved if a very careful revision is considered prior to publication. There are a couple of methodically-related points that should be made clear in advance. The referencing needs to done properly especially in the introduction and the method sections. The quality of some figures can to be further improved. I would also suggest trying extra efforts to argue for and verify some of the model features: especially the very interesting, newly-imaged low shear wave velocity zone in the lower crust across the entire Bohemian Massif that has not been imaged before. The text itself can be shortened and streamlined more. Extra spaces and minor typos are frequently occurring in the text but can be easily tracked and corrected, I gave examples in the section of the technical comments.

More specific comments

- Since the cross correlation functions are available, I am wondering why the phase velocities are not included in the tomographic inversion as well.

- In Fig 3a and b, what is the bandpass filter applied to the shown cross-correlation functions? Can you show the CCF at different filter bands? Also, in the same figure the authors show the directionality of the sources, but they did not discuss its impact on the group velocity dispersion measurements. Would be very good if they can show the seasonal variability of the dispersion curves estimated for one or two station pairs. In addition, I am wondering about how large the period-dependent signal to noise ratio

[Figure]

during the FTAN that is more relevant to the quality of the dispersion measurement rather than how noisy the cross correlation function is. I would suggest adding a figure to show the average signal to noise ratio as function of period for all station pairs instead of Fig. 3c. The variability of the signal to the noise ratio as a function of the azimuth and inter-station distance can be investigated at specific period.

- Figure 4 might be better added to the supplement, right?

- L 134 - 136: the employed tomographic method, the fast marching algorithm, allows for smoothing and regularizing the inversion. It is not clear from the text how this is affecting results of the tomographic inversion. Taking this into account, how dependent is the final group velocity maps from the initial models? Can you show a measure of model roughness and smoothness? May be an L-Curve would be of help? How do you weigh the picking errors in travel times during the inversion? Do you estimate the uncertainties of the group velocities as function of period from the tomographic inversion?

- Using a standard synthetic checkerboard tests might not be of help to effectively test the lateral resolution of the inversion or even to simulate the effects of errors in the measurements due to travel time picking procedures of the FMM. I would suggest performing more tests with adding Gaussian noise to a synthetic input model or may be at randomly distributed spikes.

- Fig. 5b: is not clear if this is a group velocity map from the inversion or just representing the path coverage at this period? Can you please indicate that also in the figure caption and explain it a bit more?

- The grid size is set to 0.1° x 0.1° during the tomographic inversion. How does the different grid sizes affect the tomography results at different periods? Important that you show examples of the final group velocity maps at different periods (from short to long periods) sampling the period range in which the measurements have been obtained and the corresponding path coverage maps at same periods.

- Is there a reason for constructing the local dispersion curves at 0.2° x 0.3° grid, while the group velocity maps have a regular 0.1 x 0.1 ° grid?

- What is the period range used for the 1-D inversion? Did you apply any quality measures and/or outliers removal to the rough parts of the dispersion curve? A primary reason for the large misfit between the measured (dashed) and the synthetic (red) dispersion curves in Fig. 6b might be due to the erroneous group velocity measurements at very short periods that are affected by the interference with higher modes. At longer periods, the group velocity measurements might not be well constrained due to the low number of crossing paths. Rejecting those parts of the dispersion curves may help improving the misfit.

- L 168 – 169: This cannot be generalized just by looking at the results at one node. Can you do the same test at different locations? Tests might be done with and without the rejection of the rough parts of the dispersion curves.

- How do you estimate the Moho depth from the surface wave inversion? How different are these estimates from those of the initial models? It is suggested to show a map of the errors in the Moho estimates from group velocities inversion. The discrepancies between Moho estimates from Receiver functions and surface wave inversion are partly due to the different sensitivities of both data sets to the Moho interface. Furthermore, RFs suffer from the inherited non-uniqueness solution (velocity-depth trade-off). It is not clear from the text how you do deal with this issue. Meanwhile, the surface wave dispersion measurements are sensitive to velocity gradients across the interfaces. In the future, the authors may wish to perform a joint inversion of the two data sets. This might help minimizing the trade-offs and also to better constrain the crustal structure and Moho topography.

- In most parts of the BM, Moho depth estimates from the Receiver function measurements are in a good agreement with those from CSS profiling. However, they do not correlate with Moho depths from the inversion of surface wave dispersion data. Can

you please comment on what could be the reason for that?

- L 376 – 380: Relying on the layer-stripping technique, the authors stated that 3-D Vs ambient noise-based measurements are much more realistic constraining the crustal structure and the sharp Moho interface than the CSS studies. However, Majdanski and Polkowski (2014) applied the same technique in the CSS data analysis and showed that the layer stripping method has its own limitations: i.e. large uncertainties are expected with increasing the number of iterations. Therefore, it is suggested that the authors show the effect of the increasing or decreasing number of layers as well as the frequency bands used in the inversion on the associated uncertainties in the resulted shear velocities as function of depth.

- Relying only on this argument, I would say that the sentences on preferring the ambient noise tomography than CSS-based results between line 376 and line 380 cannot be accepted like this and has to reformatted or even deleted as it is generally agreed, that CSS studies give the most detailed images of the crustal structures, intra-crustal interfaces and Moho depth.

- Fig. 8b and c: how do you define the thickness of the upper and the lower crustal layers? Do these velocities represent average Vs values in both layers, respectively?

- Fig. 10 shows the depth to the top of the imaged velocity-drop interface (VDI). Taking into account its undulating topography that apparently has no correlation with the Moho topography, how thick is the imaged low velocity layer? Is the thickness of this layer laterally variable? What is the vertical resolution of the Vs model? Is this feature well resolved? The trade-off between lower crust and its low velocity zone, the upper mantle velocities and the Moho depth should be checked.

- In Fig. 16. The authors argue that NW-SE shortening of the core of the BM produced or enhanced already existing horizontal foliation (fabric) within the lower crustal rocks. However, shortening would be expected to lead to vertical foliation instead. Can you please comment on that?

- The discussion presented in subsection "6.3.2 - Anisotropy of lower crust" is mainly based on the discrepancies between the high Vp velocities from CSS data available in the region and the low Vs velocities in the lower crust from the presented model. The presented evidences are not clearly supporting the presence of the anisotropic lower crust beneath the entire Bohemian Massif. Thus, I would suggest that the authors make extra efforts to reconsider this part of the discussion.

- The quality of the figures in general is also of another major concern: several figures are not decent, e.g. Fig. 5, 8, 9 and 13. Colors are misleading and barely distinguishable sometimes, for example the measured dispersion curves in Fig. 9 are hardly visible. In addition, the font-size of the labels and titles are too small.

Technical corrections

The manuscript is full of typos like the following: I suggest a carful revision and cross checking of similar issues. In the following only few examples are listed:

- In Fig. 2, the map shows two different map projections. Why? Correct it please?

- L 9: with increasing -> with the increasing

- L 14: $\sim$0.2 -> $\sim$ 0.2 space is missing

- L 28: central Europe . -> Excess space at the end of the sentence.

- L 33: References to Fig. 1 are missing.

- L 37 -38, this is very general sentence, please cite references properly. Who did what?

- L 47 - 48: similarly please cite references properly. You may indicate that, for example, Shapiro et al. (2005) inferred group velocity map at a local scale, and Yan et al. (2007) also deduced group velocity maps at a regional scale.

- L 49: is in its -> remove "in"

- L 54: remove excess spaces.

- L 59: What do you mean by Rayleigh wave extraction? Not clear!

- L 63: standard deviations, -> remove the comma

- L 69: the number of the stations is not consistent in the abstract and data section, is this on purpose?

- L 98 - 99: thanks to the . . . -> please remove or reformulate it?

- L 113: space is missing at the beginning of the sentence. . . and so on!

- vS -> should be Vs and also Vp!

- Caption of Fig. 10, circles is -> circles are — This happens frequently.

- L 206 and 207: what do the numbers (1, 7) in the subscripts refer to in the equation?

- L 370 – 371: Not clear enough, what do you mean?

―――――――――――――――――

---

## Author Comment (AC1) · 8 Jan 2021

General Comments

This is a very well written manuscript describing a new high-resolution 3D model of the shear wave velocities in the Bohemian Massif, based on ambient noise data. It contains an informative introduction, followed by short information about used data

(selected 24 258 station pairs for ambient noise processing) and detailed description of used methods (an iterative fast marching method resulting in shear velocity maps and a stochastic inversion of dispersion curves for a collection of multi-layered shear velocity models). Furter it describe the results with a large number of figures and useful supplements, resolution and sensitivity tests. The main content of the article is the discussion, that itself is a review of all important articles describing the structure of the Bohemian Massif and especially its lower crust.

The manuscript is long, but clearly describe data analysis, chosen assumptions and methods limitations. It is interesting as it presents the low shear wave velocity anomaly in the lower crust of the whole Bohemian Massif in contradiction to everything that was previously published, and explain that by anisotropy of seismic velocity. As an easy improvement, this manuscript could be shortened and even some figures (e.g. fig.5, 7, 8, 13) could be removed without harming the main message.

> *We shorten the paper by moving several figures, including the Figs. 5, 7and 13 into the supplement, but we keep the Fig.8 (Moho depth map) in the main text.*

Specific comments

1.  Reduction of shear wave velocities in the lower crust is a new and surprising result. No such effect was visible in previous P-wave velocity results from CSS experiments. It is, of course, possible that P and S waves have different characteristics, but would it be possible to verify S-wave velocities in the lower crust using existing CSS data? It would be less precise than P-waves, but still, such characteristics should be clearly visible.

> *It would be very interesting study, but such task is for specialists in CSS, having access to data and extremely large experience in CSS processing. The reduction of only $v_S$ results in an increased of vp/vs to ~1.9, which is a value typical for mafic rocks.*

The most surprising is the existence of those lower velocities in the whole area. As shown using P-wave CSS at profiles CEL09 (Hrubcova et al. 2005 JGR doi: 10.1029/2004JB003080), and S02, S03 (Majdanski et al. 2006 Tectonophysics doi: 10.1016/j.tecto.2005.10.042) anomalous P-wave velocities in form of layered lower crust were recognized in part of profiles, but not globally. Those areas were interpreter as remelting of the lower crust or magmatic underplating (Majdanski´ et al., Tectono-physics, doi: 10.1016/j.tecto.2007.02.015). Is it really a global feature of Vs of BM?

> *The ANT model shows systematically the low shear velocities for the majority of the BM. Moreover, the low velocities correlate with the tectonics of the BM (see Fig.10).*

2.  Despite the possible difference in P and S-waves velocities in the lower crust that are possible the boundary of the lower crust, both upper and lower (Moho) should match the P-wave models, at least in range of the uncertainty of used methods. As shown they are quite different. The argument of 3D modelling advantage over 2D CSS is valid, but this effect is not that strong, because in the Bohemian Massif horizontal variations are not that significant. The question is, what level of S-wave anisotropy in the upper and middle crust would be needed to match P-wave models boundaries? As shown by Sroda 2006 GRL doi:10.1029/2006GL027701, for P-wave 10% anisotropy was observed.

> *We presented in the ms. (Lines 491-491) that " the minimum strength of the lower crust anisotropy ranges from 3 to 9 % depending on the average $v_S$.*

3.  I strongly disagree that the results of CSS should be neglected as less precise comparing to ambient noise methods. From personal experience, I am convinced that controlled source seismic and P-waves analysis is the most precise method of studying the crustal structure. The second one is CSS with S-waves (as they are less precise to recognize because arrive later, are converted and mixed). The next in precision are receiver function methods, and the least precise the Surface waves (dispersion anal-ysis) methods. Authors refer to my paper (Majdanski´ and Polkowski, 2014 PAGEOPH doi 10.1007/s00024-014-0840-9) as weakness of CSS.

*We have never claimed that the results of CSS should be neglected. We compare the BM models from different approaches, without giving a preference to any of them. We are aware the three methods are sensitive to different parameters. For example, receiver function is poorer for absolute velocities than ambient noise tomography, but approaches P wave CSS for discontinuities.*

*We reformulate L378: "Therefore, the ANT method can image 3-D structures of the crust in a more realistic way compared to the case when the 2-D CSS lines do not match at crossing points."*

This paper shows the limitation of the layer-stripping approach. With previous paper (Majdanski´ Geophysics doi: 10.1190/GEO2012-0280.1) it proves that the uncertainty of layer-stripping approach grows as dv*sqrt(n), where n is the number of stripping iterations. The same is valid for presented layer-stripping ANT in the discussed paper. So what is the final uncertainty of the presented model?

*The layer-stripping in trial-and-error CSS model building method is a completely different concept from the layer-stripping in the surface wave stochastic inversion, therefore, conclusions of Majdanski and Polkowski (2014) are not applicable to the layer-stripping technique performed in this study. See also the answers RC2.23, RC2.27 and Fig.6b vs.6d.*

*The vertical resolution of the ANT vs model is depth dependent. Statistics computed on the group of the best fitting models, particularly standard deviation and skewness, can serve as a rough guide of the depth resolution and as a confirmation, that the model space in the stochastic inversion is fully explored and that the constrains of the model space do not influence the outcome. We created the map of the standard deviation and skewness for LPC/UPC boundary and for Moho (RC_Fig8, for the supplement), it show that skewness coefficients are very small values mostly around zero and standard deviations are within the generally accepted accuracy of Moho depth estimate ~5km (Kästle et al., 2018).*

Small technical corrections: *Corrected.*

The manuscript is partially written in the first person: "We picked", "We interpolate".
Should be in third person.

28: remove space

166: remove space

405: remove . .

I hope that my comments will help to improve this manuscript.

---

## Author Comment (AC2) · 8 Jan 2021

From the General comments:
RC2.1
The referencing needs to done properly especially in the introduction and the method sections.

Answer:
We complemented the references as suggested in "More specific comments" below.

RC2.2
The quality of some figures can to be further improved.

Answer
The quality of some figures was compromised by size requirements for submitting the MS, for which we have converted original images to bitmap with resolution of only 96dpi. For the final version we will supply full resolution vectorized images. The size of fonts will be adjusted in the final submission.

RC2.3
I would also suggest trying extra efforts to argue for and verify some of the model features: especially the very interesting, newly-imaged low shear wave velocity zone in the lower crust across the entire Bohemian Massif that has not been imaged before.

Answer
We modify Section "6.3.2 - Anisotropy of lower crust", for more detail see Answer RC2.42 and updated Fig. 16 (see RC2.41).

RC2.4
The text itself can be shortened and streamlined more.

Answer
We shorten the MS by moving the figures 3, 5, 7and 13 into the supplement and modify the text accordingly.

More specific comments
RC2.5
Since the cross correlation functions are available, I am wondering why the phase velocities are not included in the tomographic inversion as well.

Answer
We are aware of various techniques recently used for imaging the crust. Each has its own advantages and limitations. At this stage of the research we have decided to use the robust technique of surface wave group velocity inversion.  Future studies will investigate joint inversion of phase and group velocities, as well as the Love waves and Receiver functions, as we mentioned in the Conclusions.

RC2.6
In Fig 3a and b, what is the bandpass filter applied to the shown cross-correlation functions?

Answer
The bandpass filter in the 5-50s range was shown in the figure heading, we add the filter to the figure caption. To shorten the MS, we move Fig. 3 to the supplement.

RC2.7
Can you show the CCF at different filter bands?

Answer
To show the CCF at different filter bands, we have created a set of new images showing CCFs and noise directionality attributes with various bandpass filters. To shorten the MS as requested, the new Fig. 3 is in the supplement.

RC2.8
Also, in the same figure the authors show the directionality of the sources, but they did not discuss its impact on the group velocity dispersion measurements. Would be very good if they can show the seasonal variability of the dispersion curves estimated for one or two station pairs.

Answer

We have analysed the seasonal variations of noise sources in order to identify the best interval for ambient noise processing. Based on this analysis we have used data only from summer season which appears most isotropic in all measures. Considering the length of MS and thanks to this data selection we do expect neither any merit nor benefit for this paper by including further analysis of seasonal variations of any intermediate result from cross-correlation stage onward. Our focus is on regional structure. Furthermore, there are many papers on the subject of ambient noise sources, eg:

Stehly, L., M. Campillo, and N. M. Shapiro (2006), A study of the seismic noise from its long-range correlation properties,
J. Geophys. Res., 111, B10306, doi:10.1029/2005JB004237
Campillo, M. & Roux, Philippe. (2015). Seismic imaging and monitoring with ambient noise correlations. 1. 256-271.
Behr, Yannik & Townend, J & Bowen, Melissa & Carter, Lionel & Gorman, Richard & Brooks, Laura & Bannister, Stephen. (2013). Source directionality of ambient seismic noise inferred from three-component beamforming. Journal of Geophysical Research. 118. 240-248. 10.1029/2012JB009382.
Landès, M., F. Hubans, N. M. Shapiro, A. Paul, and M. Campillo (2010), Origin of deep ocean microseisms by using
teleseismic body waves, J. Geophys. Res., 115, B05302, doi:10.1029/2009JB006918.
Meier Ueli, Florent Brenguier, N. M. Shapiro. Detecting seasonal variations in seismic ve-locities within Los Angeles basin from correlations of ambient seismic noise.. Geophysical
Journal International, Oxford University Press (OUP), 2010, Volume 181 (Issue 2), p. 985-996.
Nishida, K., H. Kawakatsu, Y. Fukao, and K. Obara (2008), Background Love and Rayleigh waves simultaneously generated at the Pacific Ocean floors, Geophys. Res. Lett., 35, L16307, doi:10.1029/2008GL034753.
Yang, Y., and M. H. Ritzwoller (2008), Characteristics of ambient seismic noise as a source for surface wave tomography,
Geochem. Geophys. Geosyst., 9, Q02008, doi:10.1029/2007GC001814.

RC2.9
In addition, I am wondering about how large the period-dependent signal to noise ratio during the FTAN that is more relevant to the quality of the dispersion measurement rather than how noisy the cross correlation function is.
Answer
As mentioned above, we modify the original Figure 3 and in the new figure we show the signal to noise ratio as a function of period. See also our answer RC2.6 and RC2.7.

RC2.10
I would suggest adding a figure to show the average signal to noise ratio as function of period for all station pairs instead of Fig. 3c. The variability of the signal to the noise ratio as a function of the azimuth and inter-station distance can be investigated at specific period.

Answer
Instead of creating a new figure and making the MS even longer, we complement the new Fig.3 (now in the supplement) with value of the average signal-to-noise ratio for each period band (see RC_Fig3).

RC2.11
Figure 4 might be better added to the supplement, right?

Answer
We think the processing flow-chart carries the key information for the MS and thus we prefer to keep it in the main text.

RC2.12
L 134 - 136: the employed tomographic method, the fast marching algorithm, allows for smoothing and regularizing the inversion. It is not clear from the text how this is affecting results of the tomographic inversion.

Answer
We have thoroughly tested inversion parameters with the aim to obtain smooth model without appearance of footprint of station pairs, and which yet satisfies the data as recommended in the FMST software documentation http://www.iearth.org.au/codes/FMST/instructions.pdf (the 3$^{rd}$ paragraph of chapter 1 and the 1$^{st}$ paragraph of chapter 6).
We complement this information to the L138.

RC2.13
Taking this into account, how dependent is the final group velocity maps from the initial models?

Answer
In order to minimize the dependency of the group velocity model on the initial model we have tested various constant-velocity initial models, constant radius smoothing parameter and variable smoothing factor in the inversion. With such parameterization (also mentioned in L136-L138) and after six iterations, the final model is independent on the initial velocity model.

RC2.14
Can you show a measure of model roughness and smoothness? May be an L-Curve would be of help?

Answer
In the revised MS we show in the supplement new maps of final group velocities, as well as their roughness computed as RMS of the 2$^{nd}$ derivative (i.e., mean curvature) for selected periods (RC_Fig5).

RC2.15
How do you weigh the picking errors in travel times during the inversion?

Answer
Weights are computed from picking errors by an internal algorithm of the inversion program FMST (Rawlinson, 2005)

RC2.16
Do you estimate the uncertainties of the group velocities as function of period from the tomographic inversion?

Answer
Instead of uncertainty estimates of group velocities from the tomographic inversion, the FMST inversion code (Rawlinson, 2005) computes residuals of travel-times as a function of period. It is shown as a red curve in Fig 13b (now in the supplement of the revised version).

RC2.17
Using a standard synthetic checkerboard tests might not be of help to effectively test the lateral resolution of the inversion or even to simulate the effects of errors in the measurements due to travel time picking procedures of the FMM. I would suggest performing more tests with adding Gaussian noise to a synthetic input model or may be at randomly distributed spikes.

Answer
We appreciate this comment as we fully agree with this point. We tested various levels of Gaussian noise added to synthetic travel-times and tested their effects on the resolution. The tests showed that the noise level corresponding to 50% of velocity variations in the synthetic velocity model (i.e., 1.5% of synthetic travel times) does not change the general picture of the retrieved velocity heterogeneities. We replace Fig 12b by a new RC_Fig12b and update the caption.

RC2.18
Fig. 5b: is not clear if this is a group velocity map from the inversion or just representing the path coverage at this period? Can you please indicate that also in the figure caption and explain it a bit more?

Answer
Fig. 5b shows both group velocities as well as path coverage at 19 s period: the group velocities of all station pairs at period of 19 s are represented by colors of the lines and the lines represent the geometry of the path coverage.

RC2.19
The grid size is set to 0.1° x 0.1° during the tomographic inversion. How does the different grid sizes affect the tomography results at different periods?

Answer

The 2D tomography grid is intermediate processing grid, which was intentionally finer than the final 1D inversion grid. The finer grid definition at this processing stage does not affect the final 1D tomography results computed at sparser grid.

RC2.20
Important that you show examples of the final group velocity maps at different periods (from short to long periods) sampling the period range in which the measurements have been obtained and the corresponding path coverage maps at same periods.

Answer
We understand this sentence that we are requested to show the final group velocity maps at different periods, which we include in the new figure in the supplement RC_Fig5. We show these for five different periods, along with roughness (see answer to RC2.14). Sampling the period range in which the measurements have been obtained was shown in the original Fig.5 with new updated caption. The path coverage maps were shown in the Fig12c a,d are also in the new RC_Fig12c.

RC2.21
Is there a reason for constructing the local dispersion curves at 0.2° x 0.3° grid, while the group velocity maps have a regular 0.1 x 0.1 ° grid?

Answer
The 0.2x0.3 grid is the grid of final model. The size of the final grid was estimated based on spatial resolution test results at the shorter periods. The reasoning of the grid size of group velocity maps is given in answer to RC2.19.

RC2.22
What is the period range used for the 1-D inversion?

Answer
We use the period range of 2-74s for the 1-D inversion. We update the sentence on L143 as follows:
"Then we invert the regularized dispersion curves in the full range of 2-74s"

RC2.23
Did you apply any quality measures and/or outliers removal to the rough parts of the dispersion curve?

Answer
With the layer-stripping approach in the stochastic inversion, there is no need for outlier removal as this approach progressively increases the relative depth sensitivity from short to long periods with every run of the layer stripping. As we mentioned on L162-165 of the original text, the concept of layer-stripping technique in the stochastic inversion is the relative enhancement of depth sensitivities in the deeper parts of vertical profile. Fixing thickness and velocity of shallow layers (stripping) suppresses effects of the short periods which are highly sensitive in the shallow parts as it is shown in the sensitivity kernels (see also slide 16 of https://ds.iris.edu/media/workshop/2013/01/advanced-studies-institute-on-seismological-research/files/Surface_Waves_ASI.pdf)

RC2.24
A primary reason for the large misfit between the measured (dashed) and the synthetic (red) dispersion curves in Fig. 6b might be due to the erroneous group velocity measurements at very short periods that are affected by the interference with higher modes.

Answer
Figure 6b shows the best fit results in case of the layer stripping: misfit lower than 4%, in comparison with Fig. 6a (no layer stripping and misfit exceeding 6%). Furthermore, in the L126-127 of the original MS, we mentioned that "In the short-period band, the picker prioritizes the low velocity local maxima. This prevents picking the higher modes, as well as skipping and mixing different phases at the short-period range."

RC2.25

At longer periods, the group velocity measurements might not be well constrained due to the low number of crossing paths. Rejecting those parts of the dispersion curves may help improving the misfit.

Answer
Sensitivities of the long period parts of the dispersion curves, which are less well constrained due to the low number of crossing paths, are low and do not affect the inversion results in the crust. See also the sensitivity kernels in Fig. 13b and answer RC2.23.

RC2.26
L 168 – 169: This cannot be generalized just by looking at the results at one node. Can you do the same test at different locations?

Answer
The presented node serves as an example to demonstrate the need for allowing velocity drop in the stochastic inversion parametrisation (model space) to reach lower misfit (better fit). We modify the sentence, to make clear that we present an example node.

"Figure 6b,c show an example of the inversion results, in which velocity decrease with depth (Fig. 6b) needs to be allowed to achieve better fit of the dispersion curves with a local minimum around the 20 s periods."

RC2.27
Tests might be done with and without the rejection of the rough parts of the dispersion curves.

Answer
Tests to verify the possible existence of velocity decrease in the lower crust was done as suggested, i.e. by inverting the dispersion curves in the mid period range without the "rough parts of the dispersion". The selected frame of periods 12-37s is sensitive to the lower crust and contains the local minimum. While the misfits with the truncated input are lower (new RC_Fig 6d), the models fail to fit the dispersion curves outside the truncation window. Regardless of the inversion being computed with or without layer stripping, or from full or truncated window of dispersion curves, the resulting models end up with the low velocity drop in the lower part of the crust (Fig. 6a,b,d). We modify the text and captions of Fig.6 accordingly.

RC2.28
How do you estimate the Moho depth from the surface wave inversion?

Answer
We built the final Vs model by averaging 10 % of the best-fitting models over layer indexes ($v_i$, $z_i$). We mentioned this in L153-157. The averaging over layer indexes keeps layers and interfaces in the final model as per Table 1. Therefore, the Moho discontinuity is the interface between the last sub-layer of the lower part of the crust and the first sub-layer of the uppermost mantle (see Table 1). We mentioned this in L187.

We clarify the text by adding sentence to L159:
"We build the final Vs model by averaging 10 % of the best-fitting models over layer indexes."
and by updating the sentence on L187:
"Figure 8 shows depths of the Moho discontinuity, as an interface between the last sub-layer of the lower part of the crust and the first sub-layer of the uppermost mantle (see Table 1), across which the lower crust velocities change into the upper mantle velocities."

RC2.29
How different are these estimates from those of the initial models? It is suggested to show a map of the errors in the Moho estimates from group velocities inversion.

Answer
There is no initial model with pre-defined Moho in the stochastic inversion. Instead we use the model space parameters shown in Table 1.

RC2.30
The discrepancies between Moho estimates from Receiver functions and surface wave inversion are partly due to the different sensitivities of both data sets to the Moho interface. Furthermore,

RFs suffer from the inherited non-uniqueness solution (velocity-depth trade-off). It is not clear from the text how you do deal with this issue. Meanwhile, the surface wave dispersion measurements are sensitive to velocity gradients across the interfaces.

Answer
We are aware of different sensitivities of the three methods to the velocities and interfaces. The reason is to compare models of the BM from different approaches, without giving an a priori preference to any of them. We showed (1) the RF Moho, which needs a velocity model to convert delay times: the comparison of IASP91 and ANT velocities (in suppl.6,7) gives a maximum difference of 3km.
(2) 2D CSS models are azimuth dependent and consider isotropic vp;
(3) there is no simple way to use vp/vs in an anisotropic medium for recalculation the vs to vp velocities for the RF models.

RC2.31
In the future, the authors may wish to perform a joint inversion of the two data sets. This might help minimizing the trade-offs and also to better constrain the crustal structure and Moho topography.

Answer
As recommended we also intended to do the joint inversion and mentioned that in L618-L619.

RC2.32
In most parts of the BM, Moho depth estimates from the Receiver function measurements are in a good agreement with those from CSS profiling. However, they do not correlate with Moho depths from the inversion of surface wave dispersion data. Can you please comment on what could be the reason for that?

Answer
To highlight tiny differences between the Moho depths from the three methods, we used a factor 3 vertical exaggeration. This might give an impression of poor correlation between the "Moho depths from the inversion of surface wave dispersion data" with the other two methods. However differences in Moho depth are within limits considered as a good match (e.g. Kastle, 2018). We add information about the chosen exaggerated depth scale to captions of Figs 11,14 and 15.

RC2.33
L 376 – 380: Relying on the layer-stripping technique, the authors stated that 3-D Vs ambient noise-based measurements are much more realistic constraining the crustal structure and the sharp Moho interface than the CSS studies. However, Majdanski and Polkowski (2014) applied the same technique in the CSS data analysis and showed that the layer stripping method has its own limitations: i.e. large uncertainties are expected with increasing the number of iterations. Therefore, it is suggested that the authors show the effect of the increasing or decreasing number of layers as well as the frequency bands used in the inversion on the associated uncertainties in the resulted shear velocities as function of depth.

Answer
The layer-stripping in trial-and-error CSS model building method is a completely different concept from the layer-stripping in the surface wave stochastic inversion, therefore, conclusions of Majdanski and Polkowski (2014) are not applicable to the layer-stripping technique performed in this study. See also the answer to RC2.23, RC2.27 and Fig.6b vs.6d.

RC2.34
Relying only on this argument, I would say that the sentences on preferring the ambient noise tomography than CSS-based results between line 376 and line 380 cannot be accepted like this and has to reformatted or even deleted as it is generally agreed, that CSS studies give the most detailed images of the crustal structures, intra-crustal interfaces and Moho depth.

Answer

We reformulated this part (L378): "Therefore, the ANT method can image 3-D structures of the crust in a more realistic way compared to the case when the 2-D CSS lines do not match at crossing points."

RC2.35
Fig. 8b and c: how do you define the thickness of the upper and the lower crustal layers?

Answer
The variable thickness and velocities of each layer is defined in the model space of the stochastic inversion (Table 1). See answer RC2.28.

RC2.36
Do these velocities represent average Vs values in both layers, respectively?

Answer
Yes, we mentioned that in the caption of Figure 8.

RC2.37
Fig. 10 shows the depth to the top of the imaged velocity-drop interface (VDI). Taking into account its undulating topography that apparently has no correlation with the Moho topography, how thick is the imaged low velocity layer? Is the thickness of this layer laterally variable?

Answer
We mentioned that in Section 6.2.3 and particularly in the L545-548: "there is only a weak relation between the depth of the VDI or the thickness of the LPC layer and the crustal thickness." The thickness of the imaged low-velocity layer varies from 8-13 km (see also (Figs. 10, 8a).
We complement the second sentence in the text.

RC2.38
What is the vertical resolution of the Vs model?

Answer
The vertical resolution is depth dependent. Statistics computed on the group of the best fitting models, particularly standard deviation and skewness, can serve as a rough guide of the depth resolution and as a confirmation, that the model space in the stochastic inversion is fully explored and that the constrains of the model space do not influence the outcome. We created the map of the standard deviation and skewness for LPC/UPC boundary and for Moho (RC_Fig8_depth, for the supplement), it shows that skewness coefficients are very small values mostly around zero and standard deviations are within the generally accepted accuracy of Moho depth estimate ~5km (Kästle et al., 2018).

RC2.39
Is this feature well resolved?

Answer
Besides the statistic parameters for the stochastic models and the truncation tests (RC_Fig6), the good correlation of the VDI with BM tectonics is an indication that the feature is well resolved.

RC2.40
The trade-off between lower crust and its low velocity zone, the upper mantle velocities and the Moho depth should be checked.

Answer
We have checked, that the Vs in the LPC and in the upper most mantle layers are sufficiently explored by stochastic inversion, see the new RC_Fig8_vs. therefore the outcome of the stochastic inversion is not influenced by model parameter space. As mentioned in the text L380-383, the resulting velocities below the Moho correspond with those of other tomography studies.

RC2.41

In Fig. 16. The authors argue that NW-SE shortening of the core of the BM produced or enhanced already existing horizontal foliation (fabric) within the lower crustal rocks. However, shortening would be expected to lead to vertical foliation instead. Can you please comment on that?

Answer
We have complemented this caption and revised Fig. 16 (see RC_Fig16). Fabrics in the upper and lower crust, produced by the NW-SE shortening of the core of the BM, should be different. In the rigid upper crust geologists observe inclined and sub-vertical foliations, as well as elongation of the large-scale fabrics in the NE-SW directions, i.e. perpendicular to the shortening (Žák et al., 2014). Beneath the brittle-ductile transition (see also the widely accepted "jelly sandwich" model of Burov and Watts, 2006), the flow in the lower crust is the major mechanism that deforms rock materials, channeling material transport preferably in directions perpendicular to the shortening but in the horizontal plane. Lower crustal flow is a generally accepted mechanism in geodynamics, namely during processes of the continental collision. It is documented by different authors in Tibet, central China, North America and other regions. Moreover, we document the "plastic" behavior of the plagioclase feldspar (the last but one par. of 6.2.2), one of the components of the lower crust exposed in the NK Massif shown in Fig. 1.

RC2.42
The discussion presented in subsection "6.3.2 - Anisotropy of lower crust" is mainly based on the discrepancies between the high Vp velocities from CSS data available in the region and the low Vs velocities in the lower crust from the presented model. The presented evidences are not clearly supporting the presence of the anisotropic lower crust beneath the entire Bohemian Massif. Thus, I would suggest that the authors make extra efforts to reconsider this part of the discussion.

Answer
Replying to this comment, we refer to the argumentation presented in the previous paragraph and all the seismological arguments above supporting the robustness of the results. The transverse isotropy in the lower crust can explain the high P velocities observed by CSS profiling, as well as the relatively low shear-wave velocities derived by our model. This is a strong argument for our model that is supported by laboratory measurements on real rock samples. Of course, our model of the transversely isotropic lower crust is one of the possible models that geophysicists provide to the international community as an approximation of real crust. However, several decades of investigations of seismic anisotropy document that anisotropic medium in the crust and lithosphere, composed of anisotropic minerals and rocks that are exposed to stress fields, is more natural than an isotropic medium (see abundant literature in the '80s, e.g. Mereu RF, Müller S, Fountain DM (eds) Properties and processes of Earth's lower crust. AGU Geophys Mono 51/IUGG 6:121-125). This is being documented by the more and more sophisticated seismological methods and detailed observations in more recent times.

RC2.43
The quality of the figures in general is also of another major concern: several figures are not decent, e.g. Fig. 5, 8, 9 and 13. Colours are misleading and barely distinguishable sometimes, for example the measured dispersion curves in Fig. 9 are hardly visible. In addition, the font-size of the labels and titles are too small.

Answer
As mentioned above, for the final submission we will provide the high-quality plots. See the answer RC2.2

***Technical corrections***
RC2.44
In Fig. 2, the map shows two different map projections. Why? Correct it please?

Answer
All map views were generated in the same map projection.

RC2.45
L 9: with increasing -> with the increasing

L 14: ~0.2 -> ~ 0.2 space is missing
L 28: central Europe . -> Excess space at the end of the sentence.

Answer
Done

RC2.46
L 33: References to Fig. 1 are missing.

Answer
Reference to Fig. 1 was in the L33.

RC2.47
L 37 -38, this is very general sentence, please cite references properly. Who did what?

Answer
There are too many references in paper Karousova et al. (2012) to be reported here, therefore, we keep the more general referring style.

RC2.48
L 47 - 48: similarly please cite references properly. You may indicate that, for example, Shapiro et al. (2005) inferred group velocity map at a local scale, and Yan et al. (2007) also deduced group velocity maps at a regional scale.

Answer
We modify the text relating to the methodologic references as suggested.

RC2.49
L 49: is in its -> remove "in"
L 54: remove excess spaces.

Answer
Done

RC2.50
L 59: What do you mean by Rayleigh wave extraction? Not clear!

Answer
We explained this in the L119

RC2.51
L 63: standard deviations, -> remove the comma

Answer
Done

RC2.52
L 69: the number of the stations is not consistent in the abstract and data section, is this on purpose?

Answer
Numbers were consistent, in the L13 we said "more than 400 stations" and in the L69 "from 410 stations"

RC2.53
L 98 - 99: thanks to the . . . -> please remove or reformulate it?

Answer
Reformulated "Due to"

RC2.54

L 113: space is missing at the beginning of the sentence. . . and so on!
vS -> should be Vs and also Vp!
Caption of Fig. 10, circles is -> circles are — This happens frequently.

Answer
Done

RC2.55
L 206 and 207: what do the numbers (1, 7) in the subscripts refer to in the equation?

Answer
We mentioned this in the L207 "where indexes denote respective sub-layers (Table 1)"

RC2.56
L 370 – 371: Not clear enough, what do you mean?

Answer
We clarify these sentences.
The velocity remains lower even at the bottom of the crust ($v_S$ =3.6 km s$^{-1}$). The uppermost mantle velocities are lower in the ANT model than those derived from the dispersion curves of teleseismic Rayleigh waves for the half-space by Červený et al. (1977). Waves in that model arrived from back-azimuths of ~30°, which is close to the high-velocity direction derived from body-wave propagation of regional earthquakes (Plomerová et al., 1984). The presented ANT model accounts for dispersion curves from all directions, which decrease the velocities (see Table 2).

**RC_Figure 3: Examples of cross-correlation functions (CCF) for station CZ.PRU (a) and all permanent stations in summer (left) and winter (right) months. Six different band-pass filters were applied 5-50 s, 5-7 s, 10.5-13.5 s, 17-21 s, 34-40 s, and 43-53 s. The blue area marks windows used to measure RMS amplitude of coherent signals and the red area marks windows used to measure RMS amplitude of non-coherent noise. Azimuth-interstation distance rose diagrams present signal-to-noise ratio (b) and source directivity calculated as ratios of the causal and acausal parts of the CCF signals (c) for all permanent stations in the region for winter and summer period (left and right diagrams, respectively). The CCFs and measurements in the rose diagrams which do not satisfy the minimal separation condition of the station-pair at each period band are grey and shaded, respectively.**

Band-pass filter: 5-50 s

[Figure]

**Band-pass filter: 5-7 s**

[Figure]

**Band-pass filter: 10.5-13.5 s**

Summer                                                    Winter

[Figure]

Band-pass filter: 17-21 s

[Figure]

**Band-pass filter: 34-40 s**

Summer            Winter

a)

[Figure]

b)

mean SNR = 6.679

mean SNR = 6.407

c)

**Band-pass filter: 43-53 s**

[Figure]

**RC_Figure 5: Maps of final group velocities at periods 6 s, 12 s, 19 s, 37 s, and 48 s along with maps of roughness measures computed as RMS of the 2ⁿᵈ derivative (i.e., mean curvature) of the final group velocity models at selected periods.**

[Figure]

[Figure]

[Figure]

[Figure]

[Figure]

**RC_Figure 6: Examples of dispersion curves at 14.4° E 50.7° N for velocity models computed with different parameters. In all parts of the figure, dispersions from models averaged over fixed depths are blue, those from models averaged over layer indexes (xi, zi) are red. Individual parts of the figure show dispersion curves from models calculated without (a) or with (b,c) successive layer stripping; from models, where a velocity decrease was allowed (a,b) or was not allowed (c) during the inversions and from models resulting from inversions of full-range dispersion curve (a) or dispersion curve truncated at the 12–37 s interval (d). The misfit values are computed for parts of dispersion curves in frames of 12–37 s, the color represents the way of velocity model averaging. For more details see the text. Tests of gradational against sharp Moho (e) limit thickness of the potential transitional layer to 2 km. The yellow to green field marks 10 % of the best models from 400 000 computed ones; thin black lines frame 1 % of the bes-fitting models. The first quartile, median and the third quartile display misfits between the input dispersion curves and those computed for all the final 1-D models (see part (b), red curve) after each step of layer stripping (f) in period ranges of 6–48 s and 12–37 s (blue and orange, respectively). The misfit improves linearly with each run of layer stripping.**

[Figure]

**RC_Figure 8 Depth: Maps of standard deviations and skewness coefficients for selection of 10 % of the best-fitting models computed for depths of the UPC/LPC interface and the Moho . The skewness measure of dataset was computed as the Fisher-Pearson coefficient of skewness (Zwillinger and Kokoska, 2020).**

[Figure]

**Moho**

[Figure]

**RC_Figure 8 Velocity: Maps of skewness coefficients for the selection of 10 % of the best fitting models computed for the lower part of the crust (LPC) the uppermost mantle velocities.The skewness measure of dataset was computed as the Fisher-Pearson coefficient of skewness (Zwillinger and Kokoska, 2020).**

[Figure]

**RC_Figure 12: Checkerboard tests (b) with Gaussian noise added to the synthetic travel times calculated for models (a) with cell sizes of 0.6° E-W by 0.4° N-S, 0.9° E-W by 0.6° N-S and 1.2° E-W by 0.8° N-S for ray path coverage at periods 6 s, 19 s, and 48 s. Hit-counts (c) as numbers of rays crossing each cell centred around the 1–D inversion grid of 0.3° E-W by 0.2° N-S. The level of Gaussian noise added to the synthetic travel times corresponds to 50% of velocity variations in the synthetic velocity models (a).**

[Figure]

**RC_Figure 16: Scheme of tectonic processes affecting the lower-crust fabric. (a) Bottom-driven NW–SE shortening of the core of the BM produced or enhanced already existing sub-horizontal foliation (fabric) within the lower-crust rocks. (b) Late-Variscan strike-slip fault systems often developed along the pre-existing deep sutures cutting the whole lithosphere. Movement along the sutures might locally modify, overprint or erase the sub-horizontal fabric of the BM lower crust.**

[Figure]

NW-SE
shortening
of the BM

Anisotropic
fabric in the
lower crust

ST unit *suture* TB unit *suture* MD unit

Upper/Middle Crust

Lower Crust
Moho
NW-SE
shortening
of the BM

Top-most Mantle Lithosphere

Asthenosphere Flow

[Figure]

Strike-slip
faults

ST unit *suture* TB unit *suture* MD unit

Upper/ /Middle Crust

Moho

Top -most Mantle Lithosphere

Asthenosphere Flow

Destroyed anisotropy in
the lower crust
along block boundaries

---

## Author Response (AR1)

Dear Mariusz Majdanski,

Thank you for your valuable comments. We have addressed all items you have raised in the uploaded supplement. We use italic and brown color in our responses to distinguish from your comments.

Sincerely, Jiri Kvapil (on behalf of all co-authors)

Please also note the supplement to this comment:

[Figure]

https://se.copernicus.org/preprints/se-2020-176/se-2020-176-AC1-supplement.pdf

[Figure]

[Figure]
Thank you for your valuable comments. We have addressed all items you have raised in the uploaded supplement.

Sincerely, Jiri Kvapil (on behalf of all co-authors)

Please also note the supplement to this comment:
https://se.copernicus.org/preprints/se-2020-176/se-2020-176-AC2-supplement.pdf

---

## Author Response (AR2)

Dear Charlotte

We have corrected the manuscript according to Emanuel's suggestions.

Unfortunately, we have not found SE guidelines on how to submit captions for supplementary figures. We have included these captions in the zip archive alongside with supplementary figures. We hope they can be matched this way or let us know if you have different requirement.

Best Regards

Jiri